# BDC-CLIP: Brownian Distance Covariance for Adapting CLIP to Action Recognition

**Fei Long** [* 1]  **Xiaoou Li** [* 2]  **Jiaming Lv** [* 1]  **Haoyuan Yang** [1]  **Xianjun Cheng** [1]  **Peihua Li** [1]

## Abstract

Bridging contrastive language-image pre-training (CLIP) to video action recognition has attracted growing interest. Human actions are inherently rich in spatial and temporal contexts, involving dynamic interactions among people, objects, and the environment. Accurately recognizing actions requires effectively capturing these fine-grained elements and modeling their relationships with language. However, most existing methods rely on cosine similarity–practically equivalent to the Pearson correlation coefficient–between global tokens for video-language alignment. As a result, they have limited capacity to model complex dependencies and tend to overlook local tokens that encode critical spatio-temporal cues. To overcome these limitations, we propose BDC-CLIP, a novel framework that leverages Brownian Distance Covariance (BDC) to align visual and textual representations. Our method can capture complex relationships–both linear and nonlinear–between all visual and textual tokens, enabling fine-grained modeling in space, time, and language. BDC-CLIP achieves state-of-the-art performance across zero-shot, few-shot, base-to-novel, and fully supervised action recognition settings, demonstrating its effectiveness and broad applicability.

## 1. Introduction

Multimodal foundation models, such as contrastive language-image pre-training (CLIP) (Radford et al., 2021), have demonstrated remarkable zero-shot and few-shot

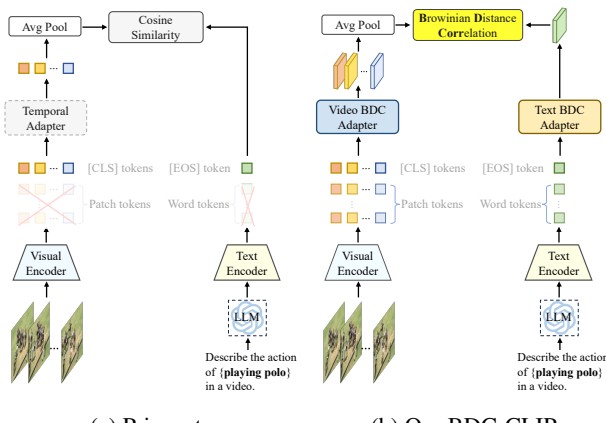

(a) Prior arts.      (b) Our BDC-CLIP.

*Figure 1.* Key differences between prior arts (a) and our BDC-CLIP (b). Previous methods align video and language based on cosine similarity between average of frame-level [CLS] tokens in video and sentence-level [EOS] token. This limits the alignment to coarse semantic matching. In contrast, BDC-CLIP aligns the two modalities via Brownian distance correlation using all visual tokens and all textual tokens, which can capture fine-grained spatio-temporal cues crucial for action recognition.

recognition capabilities. In particular, adapting CLIP to *image-centric* downstream tasks has driven significant advances (Gu et al., 2022; Zhou et al., 2022). Motivated by this success, researchers have turned their attention to similarly powerful models for video-language alignment. However, training such models from scratch remains impractical (Ni et al., 2022; Rasheed et al., 2023) due to the limited availability of large-scale video-text datasets and the prohibitive computational costs of video processing. Consequently, recent efforts focus on repurposing image-pretrained CLIP models for *video-centric* tasks, yielding promising progress (Momeni et al., 2023; Huang et al., 2024; Kim et al., 2024).

Human actions exhibit rich contextual information when captured in video and described by human language, involving dynamic interactions among people, objects, and the environment (Rasheed et al., 2023; Chen et al., 2024a). For example, the action {*playing polo*} features players swinging mallets on horseback, galloping across a grass field, with the scene continuously evolving (see Figure 5).

---

*These authors contributed equally to this work. [1] School of Information and Communication Engineering, Dalian University of Technology, Dalian, China. [2] School of Artificial Intelligence, Beijing University of Posts and Telecommunications, Beijing, China. Correspondence to: Peihua Li <peihuali@dlut.edu.cn>.

*Proceedings of the 42nd International Conference on Machine Learning*, Vancouver, Canada. PMLR 267, 2025. Copyright 2025 by the author(s).

These fine-grained elements, which are encoded as image regions within frames (Dosovitskiy et al., 2021; Caron et al., 2021) and key words in textual descriptions, are mapped via CLIP to a shared embedding space alongside corresponding textual descriptions. Effectively capturing and distinguishing these elements while modeling their relationships with language is crucial for robust video action recognition.

However, most existing methods suffer from two primary limitations in modeling video-language relations. First, they rely on cosine similarity to align video and language. Since cosine similarity is practically equivalent to the Pearson correlation coefficient (Zhelezniak et al., 2019), it can only capture linear relationships and fails to model more complex, nonlinear dependencies. Second, these methods typically focus on *global tokens*—that is, the average of frame-level [CLS] tokens for video and the sentence-level [EOS] token for text—while neglecting *local tokens* (i.e., patch tokens and word tokens) that encode fine-grained elements in video frames and textual descriptions. As a result, these approaches are limited to coarse semantic matching and lack the capacity to capture the detailed spatio-temporal cues that are essential for understanding actions in videos.

To address these limitations, we propose *BDC-CLIP*, a novel framework for video-language alignment based on *Brownian distance covariance* (BDC) (Székely & Rizzo, 2009). BDC overcomes the limitations of existing methods in two key ways. First, unlike cosine similarity, BDC can capture both linear and nonlinear correlations, enabling it to model the complex dependencies between video and language embeddings. Second, BDC naturally models relationships between *sets* of embeddings by treating them as random vectors, allowing it to fully leverage both *global tokens* and *local tokens*. This makes BDC particularly well-suited for fine-grained video-language alignment, where understanding the relationships between detailed visual and linguistic elements is essential.

Figure 1 illustrates the key differences between our approach and prior methods. BDC-CLIP introduces two core components: a video BDC adapter and a text BDC adapter. In the video BDC adapter, we leverage all visual tokens (i.e., [CLS] and patch tokens) to compute a BDC matrix as a frame-wise representation and design a temporal attention mechanism to model frame-to-frame dynamics. The video representation is then obtained by averaging these frame-wise features. On the textual side, we exploit all textual tokens (i.e., [EOS] and word tokens) to compute a BDC matrix as the text representation. Finally, we align the video and text representations using Brownian distance correlation (BDCorr), a normalized metric that is invariant to orthogonal, translational, and scaling transformations.

The primary contributions of this paper are summarized as follows:

- We introduce Brownian Distance Covariance (BDC) for multimodal alignment in foundation models such as CLIP, going beyond the limitations of cosine similarity. BDC enables the model to capture complex statistical dependencies in the video-language embedding space.

- We propose a temporal BDC attention that captures patch-wise importance and temporal dynamics, along with a language-side BDC representation derived from all textual tokens. This enables fine-grained multimodal context modeling across space, time, and language.

- Our method achieves strong performance across a range of video recognition tasks, including zero-shot, few-shot, base-to-novel, and fully supervised recognition, demonstrating its ability to capture subtle spatio-temporal cues critical for video action understanding.

## 2. Related Works

Recently many methods have been proposed for adapting image-pretrained CLIP to action recognition. They can be roughly grouped into two categories: the methods based on frozen CLIP encoders and those fine-tuning CLIP encoders.

**Adapting frozen (❄) CLIP encoders**  This research direction focuses on designing learnable prompts or adapters attached to frozen CLIP encoders. Vita-CLIP (Wasim et al., 2023) learns frame-level and video-level prompts alongside a summary video prompt for the vision encoder, as well as linguistic prompts for the text encoder. ST-Adapter (Pan et al., 2022) plugs light spatial-temporal adapters into the visual encoder. DUALPATH (Park et al., 2023) extends the visual encoder with a dual-branch architecture and introduces lightweight adapters to perform spatial and temporal modeling independently. CAST (Lee et al., 2023) also uses a two-stream architecture, with the CLIP visual encoder as the spatial expert and VideoMAE (Tong et al., 2022) as the temporal expert. It introduces a bottleneck cross-attention to enable interaction between the spatial and temporal streams. However, all these methods require backpropagation (BP) through entire encoders, making them training-inefficient. To improve efficiency, BP-free methods have been proposed. A5 (Ju et al., 2022) connects frame-level temporal adapter to the visual encoder, and meanwhile learns textual prompts. EVL (Lin et al., 2022) designs an efficient transformer decoder connected to visual encoders for capturing temporal cues. DiST (Qing et al., 2023) disentangles temporal learning via a light temporal encoder and spatial learning via CLIP encoder. MoTED (Qing et al., 2023) aligns the spatial embedding output by the frozen visual encoder with text description of categories, and also aligns temporal embeddings output by the light motion encoder with motion descriptions of categories. RIVA (Qian et al., 2024) employs slot attention to extract compact object tokens from frozen

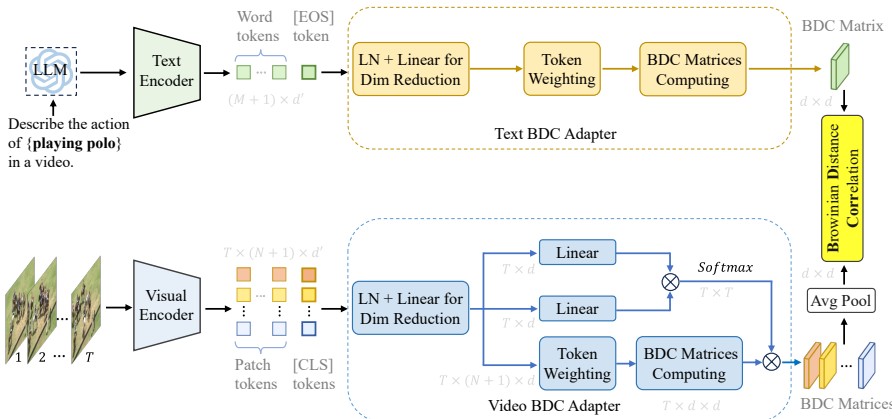

*Figure 2.* Illustration of BDC-CLIP adapting image-pretrained CLIP for action recognition in videos. Our key idea is Brownian Distance Covariance (BDC) for video-language alignment. Specifically, we develop a video BDC adapter that models temporal relation of frame-wise BDC matrices computed from weighted visual tokens. Similarly, we introduce a text BDC adapter that computes BDC matrix using weighted textual tokes as language representation. To decrease computations, we use a linear layer for dimension (dim) reduction before the BDC adapters. As in previous arts, we use LLMs to augment simple category names. Besides the BDC alignment, we also introduce a separate vision classifier (not shown for simplicity), attached to the video BDC adapter for further improvement.

pretrained models and performs temporal reasoning through object-time interactions.

**Fine-tuning (🔥) CLIP encoders**  This category focuses on fine-tuning CLIP's vision and text encoders for video tasks. ViFi-CLIP (Rasheed et al., 2023) simply finetunes both encoders with minimal design changes. It is efficient and strong, even better than previous methods with specific adapters to model temporal cues, e.g., ActionCLIP (Wang et al., 2021) and XCLIP (Ni et al., 2022). As such, the follow-up works are often based on the fine-tuned encoders. FROSTER (Huang et al., 2024) distills the fine-tuned models with the frozen CLIP models to enhance generalization ability. OST (Chen et al., 2024a) augments action class names with LLMs into Spatio-Temporal Descriptors, which are aligned to frame-level representations based on optimal transport. TC-CLIP (Kim et al., 2024) proposes a method of temporal contextualization, which summarizes informative tokens for infusing temporal cues and also serves as video-conditional text prompts. Open-VCLIP (Weng et al., 2023) introduces regularization via weight interpolation (Ilharco et al., 2022) to balance adaptation and generalization capabilities.

**BDC for deep learning**  The concept and theory of BDC were established in (Székely & Rizzo, 2009) to measure the dependence between sets of random vectors. BDC is computationally efficient, capable of modeling both linear and non-linear correlations, and provides a complete characterization of the independence of random variables. These favorable properties have motivated researchers to explore its applications in deep learning. Zhen et al. (2022) investigate the use of BDC and partial BDC (Székely & Rizzo,

2014) in tasks such as network comparison, disentangled representation learning, and improving robustness in adversarial learning. DeepBDC (Xie et al., 2022) applies the BDC metric to measure similarities between query images and support classes for *few-shot image classification*. Similarly, BDC-Adapter (Zhang et al., 2023) utilizes a BDC-based nearest neighbor classifier for *pure vision prediction*, which is integrated with vision-language prediction for final classification. Our work is inspired by both DeepBDC and BDC-Adapter but is *distinguished by two key points:* (1) we extend the BDC metric to match two distinct modalities (i.e., vision and language), rather than limiting it to unimodal tasks (i.e., vision only); and (2) we focus on video action recognition, introducing a novel BDC attention mechanism to model temporal cues, whereas prior methods focus on image recognition that requires no temporal modeling.

## 3. From Cosine Similarity to BDC

To facilitate analysis, we first introduce some notations. Let $\widetilde{\mathbf{P}} = [\widetilde{\mathbf{p}}_0, \ldots, \widetilde{\mathbf{p}}_N] \in \mathbb{R}^{d \times (N+1)}$ denote the $N+1$ visual token embeddings of $d$ dimensions for an image, where $\widetilde{\mathbf{p}}_0$ indicates the [CLS] token and the remaining entries indicate patch tokens. Similarly, let $\widetilde{\mathbf{W}} = [\widetilde{\mathbf{w}}_0, \ldots, \widetilde{\mathbf{w}}_M] \in \mathbb{R}^{d \times (M+1)}$ denote the textual token embeddings, where $\widetilde{\mathbf{w}}_0$ indicates the [EOS] token and the rest indicate word tokens.

### 3.1. A Statistical Perspective on Cosine Similarity

As described in (Zhelezniak et al., 2019), the similarity between tokens can be interpreted from a statistical perspective. Specifically, each $d$-dimensional token embedding can be regarded as a sample of $d$ observations from a ran-

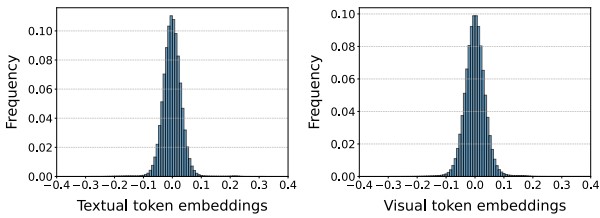

*Figure 3.* Distributions of token embeddings.

dom variable. When these samples are nearly zero-centered, cosine similarity (CS) becomes approximately equivalent to the Pearson correlation coefficient (PCC). This property holds for the CLIP model, as evidenced by the near-zero mean distributions in Figure 3.

Most existing CLIP adaptations rely on computing the CS between the [CLS] token $\widetilde{\mathbf{p}}_0$ and the [EOS] token $\widetilde{\mathbf{w}}_0$, which can be interpreted as samples drawn from two *random variables*, $\mathscr{R}_{\text{img},0}$ and $\mathscr{R}_{\text{txt},0}$, respectively. Thus, we have:

$$\texttt{CS}(\widetilde{\mathbf{p}}_0, \widetilde{\mathbf{w}}_0) \approx \texttt{PCC}(\widetilde{\mathbf{p}}_0, \widetilde{\mathbf{w}}_0) = \frac{E(\mathscr{R}_{\text{img},0}\mathscr{R}_{\text{txt},0})}{\sqrt{E(\mathscr{R}_{\text{img},0}^2)}\sqrt{E(\mathscr{R}_{\text{txt},0}^2)}}, \quad (1)$$

where $E(\cdot)$ denotes expectation over the random variable. However, PCC is limited to capturing linear relationships, and is optimal only when the joint distribution is Gaussian. It cannot capture more complex, nonlinear dependencies. As illustrated in Figure 4, the correlations between visual and textual tokens are highly nonlinear and non-Gaussian. Consequently, methods that rely solely on CS or PCC are fundamentally limited in modeling such dependencies. Moreover, these methods often ignore fine-grained information encoded in patch and word tokens across both modalities.

### 3.2. BDC for Modeling Statistical Dependency

To overcome these limitations, we propose to use BDC to match vision and language modalities by considering all tokens jointly. We treat each visual (resp. textual) embedding $\widetilde{\mathbf{p}}_i$ (resp. $\widetilde{\mathbf{w}}_j$) as a sample of $d$ observations drawn from a random variable $\mathscr{R}_{\text{img},i}$ (resp. $\mathscr{R}_{\text{txt},j}$). We then define the *random vectors*:

$$\boldsymbol{\mathscr{R}}_{\text{img}} = [\mathscr{R}_{\text{img},0}, \ldots, \mathscr{R}_{\text{img},N}], \boldsymbol{\mathscr{R}}_{\text{txt}} = [\mathscr{R}_{\text{txt},0}, \ldots, \mathscr{R}_{\text{txt},M}].$$

As formalized in Székely & Rizzo (2009), BDC measures the discrepancy between the joint distribution of $\boldsymbol{\mathscr{R}}_{\text{img}}$ and $\boldsymbol{\mathscr{R}}_{\text{txt}}$ and the product of their marginals. For the discrete case, the BDC metric can be computed in closed form using the so-called *BDC matrices*. For the visual modality, the BDC matrix $\mathbf{B}_{\text{img}} \in \mathbb{R}^{d \times d}$ is computed as:

$$\mathbf{B}_{\text{img}} = \texttt{Bdc-M}([\widetilde{\mathbf{p}}_0, \ldots, \widetilde{\mathbf{p}}_N]), \quad (2)$$

where $\texttt{Bdc-M}(\cdot)$ computes a symmetric matrix whose $(i, j)$-th entry is the Euclidean distance between the $i$-th and $j$-th rows of $\widetilde{\mathbf{P}}$, adjusted by subtracting row mean, column mean, and the global mean. The BDC matrix for textual tokens, $\mathbf{B}_{\text{txt}}$, is computed analogously. The BDC metric is then given by:

$$\texttt{BDC}(\mathbf{B}_{\text{img}}, \mathbf{B}_{\text{txt}}) = \frac{1}{d^2}\texttt{tr}(\mathbf{B}_{\text{img}}\mathbf{B}_{\text{txt}}), \quad (3)$$

where $\texttt{tr}(\cdot)$ denotes the trace of a matrix. The BDC metric can capture arbitrary statistical dependencies–both linear and nonlinear–without requiring distributional assumptions.

In the next section, we extend this framework to align videos with textual descriptions. By modeling complex dependencies across all visual tokens (across frames) and all textual tokens, our method effectively captures fine-grained semantics such as salient regions and key words critical for video understanding. This is further supported by Grad-CLIP (Zhao et al., 2024) visualizations in Figures 5 and 7.

## 4. Proposed BDC-CLIP

The framework of BDC-CLIP, as shown in Figure 2, mainly consists of a video BDC adapter and a text BDC adapter. The two adapters output respectively video and language representations, aligned by Brownian distance correlation.

### 4.1. Video-Language Alignment with BDC Metric

Consider a video sample consisting of $T$ frames and its corresponding textual description indicating an action category. These inputs are processed by the visual and text encoders, respectively. Let $\mathbf{P}^t = [\mathbf{p}_0^t, \cdots, \mathbf{p}_N^t] \in \mathbb{R}^{d' \times (N+1)}$ be the visual token embeddings of the $t$-th frame output by the visual encoder. Let $\mathbf{W} = [\mathbf{w}_0, \cdots, \mathbf{w}_M] \in \mathbb{R}^{d' \times (M+1)}$ be the textual token embeddings from the text encoder.

**Video representation** To decrease computational cost, we introduce a linear layer for dimension reduction:

$$\widehat{\mathbf{P}}^t = \texttt{Linear}(\texttt{LN}([\mathbf{p}_0^t, \cdots, \mathbf{p}_N^t])), \quad (4)$$

where $\widehat{\mathbf{P}}^t = [\widehat{\mathbf{p}}_0^t, \widehat{\mathbf{p}}_1^t, \ldots, \widehat{\mathbf{p}}_N^t] \in \mathbb{R}^{d \times (N+1)}$, LN denotes layer normalization, and Linear denotes a linear transformation reducing the feature dimension from $d'$ to $d$ ($d < d'$).

We construct the video representation using all visual tokens per frame. Notably, different patch tokens contribute differently to video recognition tasks. For instance, certain patch tokens may capture essential parts of the subjects (e.g., persons or objects), while others may represent background or irrelevant regions. To quantify token importance, we introduce a weighting mechanism based on the similarity of each token to the [CLS] token. Since the [CLS] token is designed to encapsulate the holistic representation of a

frame, it is reasonable to assign greater importance to tokens with higher similarity to the [CLS] token. The weighted embedding of a patch token is computed as:

$$\widetilde{\mathbf{p}}_i^t = \omega_i^t \widehat{\mathbf{p}}_i^t, \quad \omega_i^t = \text{Softmax}\left(\widehat{\mathbf{p}}_0^t \cdot \widehat{\mathbf{p}}_i^t / \sqrt{d}\right), \quad (5)$$

where $\cdot$ denotes the dot product of two vectors, and $\text{Softmax}$ is the softmax function. The BDC matrix for the $t$-th frame is computed as:

$$\mathbf{B}_{\text{img}}^t = \text{Bdc-M}([\widetilde{\mathbf{p}}_0^t, \cdots, \widetilde{\mathbf{p}}_N^t]), \quad (6)$$

For efficient processing, we perform half-vectorization (Vech) for $\mathbf{B}^t$ by stacking the entries on and below its diagonals, obtaining a compact vector $\mathbf{b}_{\text{img}}^t = \text{Vech}(\mathbf{B}_{\text{img}}^t) \in \mathbb{R}^{d(d+1)/2}$.

Then, we learn temporal relationships across frames with an attention mechanism. Let the values be $\mathbf{V} = [\mathbf{b}_{\text{img}}^1, \mathbf{b}_{\text{img}}^2, \cdots, \mathbf{b}_{\text{img}}^T]$. We feed the frame-wise [CLS] tokens $[\widehat{\mathbf{p}}_0^1, \cdots, \widehat{\mathbf{p}}_0^T]$ to two separate linear layers, achieving the queries $\mathbf{Q} \in \mathbb{R}^{d \times T}$ and keys $\mathbf{K} \in \mathbb{R}^{d \times T}$, respectively. The attention module outputs:

$$\widetilde{\mathbf{V}} = \text{Attention}[\mathbf{Q}, \mathbf{K}, \mathbf{V}]. \quad (7)$$

Finally, we perform average (Avg) pooling across $T$ frames and obtain the video-level representation:

$$\mathbf{B}_{\text{vid}} = \text{Sym-h}(\text{Avg-pool}(\widetilde{\mathbf{V}})), \quad (8)$$

where $\text{Sym-h}$ reconstructs a symmetric matrix from a vector.

**Text representation** The computation for text representation mirrors that of video representation, but without temporal attention. Given textual tokens $\mathbf{W}$, we first achieve the embeddings of reduced dimension $\widehat{\mathbf{W}} \in \mathbb{R}^{d \times (M+1)}$ through a layer normalization and a linear layer, which are then weighted in terms of their similarities to the [EOS] token $\widehat{\mathbf{w}}_0$ for obtaining $\widetilde{\mathbf{w}}_i, i = 0, 1, \cdots, M$. The BDC matrix for text representation is then computed as:

$$\mathbf{B}_{\text{txt}} = \text{Bdc-M}([\widetilde{\mathbf{w}}_0, \cdots, \widetilde{\mathbf{w}}_M]). \quad (9)$$

**Vision-language alignment** The similarity between video and text representations is measured using Brownian Distance Correlation (BDCorr):

$$\text{BDCorr}(\mathbf{B}_{\text{vid}}, \mathbf{B}_{\text{txt}}) = \frac{\text{tr}(\mathbf{B}_{\text{vid}}\mathbf{B}_{\text{txt}})}{\sqrt{\text{tr}(\mathbf{B}_{\text{vid}}\mathbf{B}_{\text{vid}})}\sqrt{\text{tr}(\mathbf{B}_{\text{txt}}\mathbf{B}_{\text{txt}})}} \quad (10)$$

The values of BDCorr lie in the interval $[0, 1]$, where a value of zero indicates that the two modalities are independent of each other. It is worth mentioning that BDCorr is invariant to orthogonal, translational, and scaling transformations.

By leveraging all visual and textual tokens, BDC-CLIP captures fine-grained relationships across spatial, temporal, and linguistic domains. In contrast, cosine similarity-based methods capture only coarse, linear correlations between global tokens.

### 4.2. Training Objective

In addition to aligning video and text BDC representations, we introduce a vision classifier built upon the video BDC adapter, leveraging ground-truth labels for supervision. Consistent with prior works (Chen et al., 2024a; Zhang et al., 2024), we retain the backbone loss, i.e., CLIP's original contrastive loss, which directly aligns the embeddings from the visual and textual backbone encoders.

Let $\mathbf{B}_{\text{txt}}(c)$ be the textual BDC matrix of the $c$-th category, and let $S_{\text{BDC}}(c) = \text{BDCorr}(\mathbf{B}_{\text{vid}}, \mathbf{B}_{\text{txt}}(c))$. Additionally, let $\mathbf{y} = [y_c]_{c=1}^C$ be the ground-truth probability vector, where $C$ is the number of categories. The Vision-Language Contrastive (VL-Ctr) loss for the BDC adapters is defined as:

$$L_{\text{adapter}}^{\text{VL-Ctr}} = \sum_{c=1}^{C} y_c \log\left(\frac{\exp(S_{\text{BDC}}(c)/\tau)}{\sum_{c'=1}^{C} \exp(S_{\text{BDC}}(c')/\tau)}\right), \quad (11)$$

where $\tau$ is a learnable temperature parameter.

For the vision classifier, the half-vectorized video representation $\mathbf{B}_{\text{vid}}$ is passed through a fully-connected layer parameterized by $\mathbf{R}$ for softmax classification. The classification loss is given by:

$$L_{\text{adapter}}^{\text{V-Cls}} = \sum_{c=1}^{C} y_c \log(\text{Softmax}(\text{Vech}(\mathbf{B}_{\text{vid}})\mathbf{R})_c), \quad (12)$$

where $(\cdot)_c$ denotes the $c$-th component of the softmax output. This supervised loss encourages learning of more discriminative and generalizable features.

The backbone loss is based on cosine similarity (Rasheed et al., 2023):

$$L_{\text{backbone}}^{\text{VL-Ctr}} = \sum_{c=1}^{C} y_c \log\left(\frac{\exp(S_{\cos}(c)/\tau^*)}{\sum_{c'=1}^{C} \exp(S_{\cos}(c')/\tau^*)}\right), \quad (13)$$

where $S_{\cos}(c) = \cos(\text{Avg-pool}([\mathbf{p}_0^t]_{t=1}^T), \mathbf{w}_0(c))$, $\mathbf{w}_0(c)$ is the [EOS] token of the $c$-th category, and $\tau^*$ is a learnable temperature parameter.

The overall loss function combines the three components:

$$L_{\text{total}} = L_{\text{adapter}}^{\text{VL-Ctr}} + L_{\text{adapter}}^{\text{V-Cls}} + L_{\text{backbone}}^{\text{VL-Ctr}}. \quad (14)$$

## 5. Experiments

We first describe the experimental setup (§5.1). Then we compare to state-of-the-art methods in light of performance (§5.2) and cost (§5.3). We finally conduct ablation study (§5.4) and provide qualitative analysis (§5.5).

## 5.1. Experimental Setup

**Datasets and task setting** We conduct experiments on five widely used action recognition datasets, i.e., Kinetics-400 (K400) (Carreira & Zisserman, 2017), Kinetics-600 (K600) (Carreira et al., 2018), HMDB-51 (Kuehne et al., 2011), UCF-101 (Soomro et al., 2012) and Something Something V2 (SSv2) (Goyal et al., 2017). Following (Rasheed et al., 2023; Kim et al., 2024; Chen et al., 2024b; Li et al., 2024), we pretrain on K400 and then evaluate on downstream tasks in zero-shot, few-shot, base-to-novel generalization and fully-supervised settings. Few-shot and base-to-novel generalization tasks directly fine-tuned from CLIP are reported in *Section B.2 of Appendix*.

**Implementation** We utilize CLIP models with a ViT-B/16 visual encoder and the corresponding text encoder throughout this paper. For text augmentation, following the approach in (Huang et al., 2024; Kim et al., 2024), we query GPT-4o to generate enriched textual descriptions for each category name using the prompt: "Describe the action of {category name} in a video." Both temperature parameters $\tau$ in Eq. 11 and $\tau^*$ in Eq. 13 in the training objective are initialized by 0.07. All experiments are conducted using GeForce RTX 4090 GPUs with the PyTorch framework.

*Detailed experimental setup is provided in Section A.*

## 5.2. Comparison to State-of-The-Art Methods

**(i) Zero-shot recognition** For this task, all models are trained on K400 and then directly evaluated on downstream datasets in zero-shot manner. Note that recent works (Kim et al., 2024; Huang et al., 2024; Lin et al., 2023; Weng et al., 2023) have exploited weight-space ensembling (WSE) (Wortsman et al., 2022) to boost performance of zero-shot task; see Section A.2 for implementation details. For a fair comparison, we report the results with (w/) and without (w/o) WSE. Table 1 presents the comparison results. The results of ActionCLIP and ViFi-CLIP w/ WSE are duplicated from TC-CLIP. *For the case of w/o WSE, the top-1 accuracies of BDC-CLIP are better than the second-best ones by 1.2%, 2.9% and 1.1% in top-1 accuracy on HMDB-51, UCF-101 and K600, respectively. All methods w/ WSE improve over their counterparts without WSE, particularly on UCF-101 and K600, while our BDC-CLIP still stands out, outperforming the runners-up by 4.7%, 1.5% and 0.8% on the three datasets.* These results suggest that BDC-CLIP has better generalization ability than the competitors.

**(ii) Few-shot recognition** In this setting, all methods are first trained on K400 and subsequently are evaluated for all-way classification provided with $K$ training examples per class. The comparison results are presented in Table 2. TC-CLIP), our BDC-CLIP outperforms on HMDB-51 by more than 2.7% in each $K$-shot task, and improves on SSv2

*Table 1.* Comparison to previous methods in *zero-shot* setting using K400-pretrained models. WSE indicates Weight-Space Ensembling; CLIP Backbone Encoders (BEs) are frozen (❄) or fine-tuned (🔥). The best results are **bold** and the second-best ones are underlined. † indicates results reproduced by us.

| | Method | BEs | HMDB-51 | UCF-101 | K600 (Top-1) | K600 (Top-5) |
|---|---|---|---|---|---|---|
| w/o WSE | A5 Ju et al. | ❄ | $44.3_{\pm 2.2}$ | $69.3_{\pm 4.2}$ | $55.8_{\pm 0.7}$ | $81.4_{\pm 0.3}$ |
| | Vita-CLIP Wasim et al. | ❄ | $48.6_{\pm 0.6}$ | $75.0_{\pm 0.6}$ | $67.4_{\pm 0.5}$ | – |
| | DiST Qing et al. | ❄ | $55.4_{\pm 1.2}$ | $72.3_{\pm 0.6}$ | – | – |
| | MoTED Zhang et al. | ❄ | $\underline{58.2}_{\pm 1.1}$ | $78.3_{\pm 0.6}$ | $69.9_{\pm 0.5}$ | – |
| | ActionCLIP Wang et al. | 🔥 | $49.1_{\pm 0.4}$ | $68.0_{\pm 0.9}$ | $56.1_{\pm 0.9}$ | $83.2_{\pm 0.2}$ |
| | X-CLIP Ni et al. | 🔥 | $44.6_{\pm 5.2}$ | $72.0_{\pm 2.3}$ | $65.2_{\pm 0.4}$ | $86.1_{\pm 0.8}$ |
| | ViFi-CLIP Rasheed et al. | 🔥 | $52.3_{\pm 0.2}$ | $78.9_{\pm 1.1}$ | $70.7_{\pm 0.8}$ | $92.1_{\pm 0.3}$ |
| | TC-CLIP† Kim et al. | 🔥 | $56.8_{\pm 0.9}$ | $\underline{83.0}_{\pm 0.6}$ | $\underline{75.4}_{\pm 0.9}$ | $\underline{94.7}_{\pm 0.4}$ |
| | BDC-CLIP (Ours) | 🔥 | $\mathbf{59.4}_{\pm 0.3}$ | $\mathbf{85.9}_{\pm 0.9}$ | $\mathbf{76.5}_{\pm 0.8}$ | $\mathbf{95.0}_{\pm 0.3}$ |
| w/ WSE | ActionCLIP Wang et al. | 🔥 | $51.9_{\pm 0.5}$ | $74.2_{\pm 1.0}$ | $67.5_{\pm 1.2}$ | $90.7_{\pm 0.1}$ |
| | ViFi-CLIP Rasheed et al. | 🔥 | $52.2_{\pm 0.7}$ | $81.0_{\pm 0.9}$ | $73.9_{\pm 0.5}$ | $93.3_{\pm 0.3}$ |
| | Open-VCLIP Weng et al. | 🔥 | $53.9_{\pm 1.2}$ | $83.4_{\pm 1.2}$ | $73.0_{\pm 0.8}$ | $93.2_{\pm 0.1}$ |
| | MAXI Lin et al. | 🔥 | $52.3_{\pm 0.7}$ | $78.2_{\pm 0.8}$ | $71.5_{\pm 0.8}$ | $92.5_{\pm 0.4}$ |
| | OST Chen et al. | 🔥 | $55.9_{\pm 1.2}$ | $79.7_{\pm 1.1}$ | $75.1_{\pm 0.6}$ | $94.6_{\pm 0.2}$ |
| | FROSTER Huang et al. | 🔥 | $54.8_{\pm 1.3}$ | $84.8_{\pm 1.1}$ | $74.8_{\pm 0.9}$ | – |
| | TC-CLIP Kim et al. | 🔥 | $\underline{56.0}_{\pm 0.3}$ | $85.4_{\pm 0.8}$ | $\underline{78.1}_{\pm 1.0}$ | $\underline{95.7}_{\pm 0.3}$ |
| | BDC-CLIP (Ours) | 🔥 | $\mathbf{60.7}_{\pm 0.5}$ | $\mathbf{86.9}_{\pm 0.9}$ | $\mathbf{78.9}_{\pm 0.8}$ | $\mathbf{95.9}_{\pm 0.2}$ |

by 1.2%, 1.5%, 2.3% and 4.3% for 2-, 4-, 8- and 16- shot tasks, while performing better on UCF-101 (~1.0%) in each of $K$-shot setting. We provide results without pretraining on K400 (i.e., directly finetuned from CLIP) in Table 8.

**(iii) Base-to-novel generalization** All methods pretrained on K400 are fine-tuned and tested on base classes, along with evaluation on novel classes in zero-shot manner. Table 3 shows Top-1 accuracies for base and novel classes and their harmonic mean (HM). We can see that BDC-CLIP achieves state-of-the-art performance across all three datasets, with HM exceeding TC-CLIP by 2.6% on HMDB-51, 2.0% on UCF-101, and 2.1% on SSv2. See Appendix Table 9 for the results without pretraining on K400.

**(iv) Fully-supervised recognition** Following the common practice for closed-set setting, we conduct experiment with standard splits on K400. As in ViFi-CLIP, we sample 16 frames per video clip and conduct inference with 4 video clips and 3 spatial crops (4×3 views). We also apply WSE to this setting. From Table 4, we can see that, without WSE, BDC-CLIP surpasses the previous best performer (TC-CLIP) by 0.4% in top-1 accuracy. Moreover, BDC-CLIP with WSE further improves, achieving a top-1 accuracy of 86.5%.

## 5.3. Computational Cost Analysis

Following TC-CLIP, we analyze the computational cost in terms of parameters, GFLOPS, and throughput (per view), as reported in Table 5. All metrics are normalized to the baseline (ViFi-CLIP), and measured on a single GeForce RTX 4090 GPU. We separately list the costs for BDC-

| Method | BEs | HMDB-51 | | | | UCF-101 | | | | SSv2 | | | |
|---|---|---|---|---|---|---|---|---|---|---|---|---|---|
| | | $K=2$ | $K=4$ | $K=8$ | $K=16$ | $K=2$ | $K=4$ | $K=8$ | $K=16$ | $K=2$ | $K=4$ | $K=8$ | $K=16$ |
| CLIP Radford et al. | ❄ | 41.9 | 41.9 | 41.9 | 41.9 | 63.6 | 63.6 | 63.6 | 63.6 | 2.7 | 2.7 | 2.7 | 2.7 |
| A5 Ju et al. | ❄ | 46.7 | 50.4 | 61.3 | 65.8 | 76.3 | 84.4 | 90.7 | 93.0 | 4.5 | 6.7 | 7.2 | 9.5 |
| ActionCLIP Wang et al. | 🔥 | 54.3 | 56.2 | 59.3 | 66.1 | 76.7 | 80.4 | 87.6 | 91.8 | 4.8 | 6.9 | 9.1 | 12.3 |
| X-CLIP Ni et al. | 🔥 | 60.5 | 66.8 | 69.3 | 71.7 | 89.0 | 91.4 | 94.7 | 96.3 | 6.6 | 7.8 | 9.9 | 13.7 |
| ViFi-CLIP Rasheed et al. | 🔥 | 63.0 | 65.1 | 69.6 | 72.0 | 91.0 | 93.7 | 95.0 | 96.4 | 6.7 | 7.9 | 10.2 | 13.5 |
| OST Chen et al. | 🔥 | 64.8 | 66.7 | 69.2 | 71.6 | 90.3 | 92.6 | 94.4 | 96.2 | 8.0 | 8.9 | 10.5 | 12.6 |
| ALT Chen et al. | 🔥 | 64.3 | 66.7 | 70.4 | 74.5 | 93.2 | 95.3 | 96.4 | 97.3 | 6.6 | 7.7 | 9.4 | 12.9 |
| TC-CLIP Kim et al. | 🔥 | 65.3 | 68.5 | 71.4 | 73.0 | 94.1 | 95.6 | 96.6 | 97.3 | 8.7 | 10.1 | 12.1 | 15.2 |
| BDC-CLIP (Ours) | 🔥 | **68.0** | **71.5** | **75.9** | **77.3** | **94.9** | **96.7** | **97.5** | **98.5** | **9.9** | **11.6** | **14.4** | **19.5** |

*Table 2.* Comparison to previous methods in *all-way K‑shot* setting with pretraining on K400. CLIP Backbone Encoders (BEs) are frozen (❄) or finetuned (🔥). The best results are **bold** and the second-best ones are underlined. *The results of directly fine-tuning from CLIP are provided in Table 8.*

*Table 3.* Comparison to previous methods in *base-to-novel generalization* setting with pretraining on K400. HM indicates the harmonic mean. *The results of directly fine-tuning from CLIP are given in Table 9.*

| Method | BEs | HMDB-51 | | | UCF-101 | | | SSv2 | | |
|---|---|---|---|---|---|---|---|---|---|---|
| | | Base | Novel | HM | Base | Novel | HM | Base | Novel | HM |
| CLIP Radford et al. | ❄ | 53.3 | 46.8 | 49.8 | 78.5 | 63.6 | 70.3 | 4.9 | 5.3 | 5.1 |
| A5 Ju et al. | ❄ | 70.4 | 51.7 | 59.6 | 95.8 | 71.0 | 81.6 | 12.9 | 5.7 | 7.9 |
| ActionCLIP Wang et al. | 🔥 | 69.0 | 57.2 | 62.6 | 85.6 | 75.3 | 80.1 | 8.1 | 8.7 | 8.4 |
| X-CLIP Ni et al. | 🔥 | 75.8 | 52.0 | 61.7 | 95.4 | 74.0 | 83.4 | 14.2 | 11.0 | 12.4 |
| ViFi-CLIP Rasheed et al. | 🔥 | 77.1 | 54.9 | 64.1 | 95.9 | 74.1 | 83.6 | 15.8 | 11.5 | 13.3 |
| TC-CLIP Kim et al. | 🔥 | 79.4 | 58.3 | 67.2 | 97.5 | 84.5 | 90.5 | 19.6 | 15.6 | 17.4 |
| BDC-CLIP (Ours) | 🔥 | **81.0** | **61.3** | **69.8** | **97.5** | **88.0** | **92.5** | **20.9** | **18.2** | **19.5** |

*Table 4.* Comparison to previous methods in *fully-supervised* setting on K400. [†]: result with WSE.

| Method | BEs | Views | Frames | Top-1 | Top-5 |
|---|---|---|---|---|---|
| DiST Qing et al. | ❄ | $3 \times 1$ | 32 | 85.0 | 97.0 |
| MoTED Zhang et al. | ❄ | $3 \times 1$ | 32 | 86.2 | 97.5 |
| AIM Yang et al. | ❄ | $3 \times 1$ | 32 | 84.7 | 96.7 |
| ALT Chen et al. | 🔥 | $3 \times 1$ | 32 | 85.5 | 96.7 |
| Vita-CLIP Wasim et al. | ❄ | $4 \times 3$ | 16 | 82.9 | 96.3 |
| X-CLIP Ni et al. | 🔥 | $4 \times 3$ | 16 | 84.7 | 96.8 |
| ViFi-CLIP Rasheed et al. | 🔥 | $4 \times 3$ | 16 | 83.9 | 96.3 |
| TC-CLIP Kim et al. | 🔥 | $4 \times 3$ | 16 | 85.2 | 96.9 |
| BDC-CLIP (Ours) | 🔥 | $4 \times 3$ | 16 | 85.6 | 96.9 |
| TC-CLIP[†] Kim et al. | 🔥 | $4 \times 3$ | 16 | 85.7 | 97.1 |
| BDC-CLIP[†] (Ours) | 🔥 | $4 \times 3$ | 16 | **86.5** | **97.4** |

*Table 5.* Comparison of *computational cost* to previous methods. Throughput is measured on a single GeForce RTX 4090 GPU.

| Method | Params (M) | GFLOPS | Throughput |
|---|---|---|---|
| ViFi-CLIP Rasheed et al. | **124.3** 1.00× | **285** 1.00× | **46** 1.00× |
| ActionCLIP Wang et al. | 143.7 1.16× | 567 1.99× | 28 0.61× |
| X-CLIP Ni et al. | 169.7 1.37× | 288 1.01× | 42 0.91× |
| Vita-CLIP Wasim et al. | 161.8 1.30× | 307 1.08× | 34 0.74× |
| OST Chen et al. | **124.3** 1.00× | 287 1.01× | 42 0.91× |
| TC-CLIP Kim et al. | 127.5 1.03× | 304 1.07× | 33 0.72× |
| BDC-CLIP (ZeroS, Ours) | 126.9 1.02× | 316 1.11× | 37 0.80× |
| BDC-CLIP (FullS, Ours) | 132.0 1.06× | 316 1.11× | 37 0.80× |

CLIP in zero-shot (ZeroS) and fully supervised (FullS) settings. Notably, BDC-CLIP requires only a modest increase over ViFi-CLIP in parameters (1.02×−1.06×) and GFLOPS (1.11×), with throughput at 0.80×. Compared to TC-CLIP, a previous top-performing method, BDC-CLIP incurs slightly higher GFLOPS but achieves faster throughput, demonstrating efficient trade-offs in computational cost.

### 5.4. Ablation Analysis

To facilitate fast ablation, we pretrain on K400-tiny (Rasheed et al., 2023) where each class has 100 training videos, and evaluate zero-shot recognition on K600 and $K$‑shot ($K = 2, 16$) recognition on HMDB-51 and SSv2.

**Component analysis** Our baseline is the original CLIP video-language (VL) alignment attached to the backbone encoders (Rasheed et al., 2023). Based on this, we analyze the role of BDC VL alignment and BDC vision (V) classification, as shown in Table 6a. It can be seen that, combination of BDC VL alignment improves significantly over the baseline by 1.8% on K600, 4.1% and 3.5% for 16-shot task on HMDB-51 and SSv2, respectively. These big gains indicate our BDC alignment can more effectively mitigate the domain gap between the image and video, and have stronger ability for video-language alignment. By further integrating a separate BDC vision classifier, the performance improves non-trivially. This suggests that the additional vision supervision is helpful in learning generalized representations.

**Metric and representation** We compare different metrics and representations. For the BDC, the vanilla method is simple application of DeepBDC (Xie et al., 2022) to videos, where the average of frame-wise BDC matrices is used as a video representation. For the cosine similarity (CS), besides the scheme of global tokens (GlobalT), the local tokens (i.e., patch tokens) can be combined, where we averagely pool all patch tokens along with the global [CLS] token as per-frame representation. We also compare to bilinear pooling (Lin et al., 2015) that uses normalized second moment of all tokens as the representations and Frobenius (Frob) distance as the metric. The results are shown in Table 6b.

We first note that all BDC-based methods are superior to CS-based ones, suggesting that the BDC metric is more suitable for video-language matching. *For the CS*, the Glo-

baT+LocalT scheme is better than the scheme of single GlobalT, which indicates that usage of local tokens benefits alignment of the two modalities. *For the BDC metric*, our BDC-CLIP outperforms DeepBDC by $1.1\%-1.3\%$ for zero- or 2-shot setting and $2.1\%-3.0\%$ for 16-shot setting. In the end, we note that *bilinear pooling* is better than CS while being significantly inferior to BDC-CLIP.

**Dimension reduction (DR)** The size of BDC matrices is quadratic of the dimension of token embeddings. To decrease computations, we introduce a linear layer for DR before the BDC adapters. We assess performance as a function of the dimension $d$. As seen in Table 6c, on the whole the accuracies increase consistently as the dimension grows until $d = 320$ and then decrease. To trade off the performance and cost, we adopt $d = 192$ across the paper.

**Text augmentation** We compare the vanilla prompt template hand-engineered by CLIP, i.e., "A video of {}", with the prompts generated by LLM. From Table 6d, we can see that LLM enhanced prompts improve non-trivially over the vanilla prompts by $0.3\%-0.7\%$ for different settings across the datasets. As such, throughout the paper we adopt the text augmentation via LLM for the category name, unless otherwise specified.

*Additional ablation study is provided in Section B.1.*

## 5.5. Qualitative Analysis

**Why cosine similarity is insufficient** Most CLIP adaptation methods (e.g., Rasheed *et al.*, 2023; Kim *et al.*, 2024) rely on cosine similarity (CS)–empirically equivalent to the Pearson correlation coefficient (PCC)–for cross-modal matching. However, since PCC only measures *linear* dependence, it cannot capture the higher-order, non-monotonic relations that often arise between vision and language. To investigate this, we present scatterplots and density contours for some representative visual-textual token pairs in Figure 4. We observe that cross-modal token relationships are non-linear and the joint distributions are non-Gaussian. In such scenarios, CS collapses disparate patterns to similar scores, whereas BDC cleanly separates matched from mismatched pairs. Recall that in Table 6b we give the dataset-level, apples-to-apples accuracy gains of BDC over CS.

**Heatmap visualization of video-text pairs** To enhance understanding, we visualize attention maps for {*playing polo*} from the K600 validation set. The models are evaluated in zero-shot manner using augmented text prompts generated by LLMs for better fine-grained alignment. Grad-ECLIP (Zhao et al., 2024) is employed to visualize the attention maps of both visual and textual modalities. As shown in Figure 5, BDC-CLIP demonstrates a stronger focus on critical regions in the dynamic frames, such as the horse and players. In the meantime, it attends more effec-

*Table 6.* Ablation analysis of BDC-CLIP.

(a) Role of components.

| | K600 | HMDB-51 | | SSv2 | |
| --- | --- | --- | --- | --- | --- |
| | Zero-shot | 2-shot | 16-shot | 2-shot | 16-shot |
| Backbone vision-language align | $71.7_{\pm 0.9}$ | 63.2 | 69.3 | 7.7 | 12.9 |
| +BDC vision-language align | $73.5_{\pm 0.7}$ | 65.9 | 73.4 | 8.7 | 16.4 |
| +BDC vision classifier | $73.8_{\pm 0.8}$ | 66.1 | 73.9 | 8.9 | 16.8 |

(b) Effect of metric and representation.

| Metric | Representation | K600 | HMDB-51 | | SSv2 | |
| --- | --- | --- | --- | --- | --- | --- |
| | | Zero-shot | 2-shot | 16-shot | 2-shot | 16-shot |
| BDC | DeepBDC [Xie et al.] | $72.5_{\pm 0.7}$ | 64.9 | 71.8 | 7.8 | 13.8 |
| | BDC-CLIP | $73.8_{\pm 0.8}$ | 66.1 | 73.9 | 8.9 | 16.8 |
| CS | GlobalT | $71.7_{\pm 0.9}$ | 63.2 | 69.3 | 7.7 | 12.9 |
| | GlobalT+LocalT | $71.8_{\pm 0.8}$ | 63.6 | 69.9 | 7.7 | 13.5 |
| Frob | Bilinear [Lin et al.] | $72.1_{\pm 0.8}$ | 63.9 | 71.2 | 7.6 | 13.5 |

(c) Effect of dimension (dim).

| Dim $d$ | K600 | HMDB-51 | | SSv2 | |
| --- | --- | --- | --- | --- | --- |
| | Zero-shot | 2-shot | 16-shot | 2-shot | 16-shot |
| 128 | $72.7_{\pm 0.8}$ | 65.7 | 73.3 | 8.4 | 17.0 |
| 192 | $73.8_{\pm 0.9}$ | 66.1 | 73.9 | 8.9 | 16.8 |
| 256 | $73.7_{\pm 0.8}$ | 66.3 | 73.7 | 8.4 | 17.1 |
| 320 | $73.8_{\pm 0.9}$ | 67.1 | 74.5 | 8.8 | 17.1 |
| 384 | $73.9_{\pm 0.7}$ | 65.7 | 73.2 | 8.7 | 16.9 |

(d) Influence of text augmentation.

| Prompts | K600 | HMDB-51 | | SSv2 | |
| --- | --- | --- | --- | --- | --- |
| | Zero-shot | 2-shot | 16-shot | 2-shot | 16-shot |
| Vanilla | $73.8_{\pm 0.8}$ | 66.1 | 73.9 | 8.9 | 16.8 |
| LLM | $74.5_{\pm 1.0}$ | 66.6 | 74.6 | 9.2 | 17.3 |

tively to the key words including 'players' and 'mallets'. This confirms that BDC-CLIP captures fine-grained cues across space, time, and language, leading to better action recognition.

*Section B.4 provides t-SNE visualization of video-language representation and additional heatmap visualizations.*

## 6. Conclusion

We propose BDC-CLIP, a novel framework for adapting image-pretrained CLIP models to video action recognition. By leveraging Brownian Distance Covariance (BDC) and utilizing all visual and textual tokens, our approach captures complex dependencies between video and language in high-dimensional embedding space. This allows BDC-CLIP to effectively exploit fine-grained cues–such as salient regions in video frames and key words in textual descriptions–that are crucial for accurate action recognition. In doing so, it overcomes the limitations of prior methods that rely solely on classical cosine similarity. The strong performance of BDC-CLIP across a range of action recognition benchmarks highlights the value of advanced statistical metrics like BDC for multimodal learning.

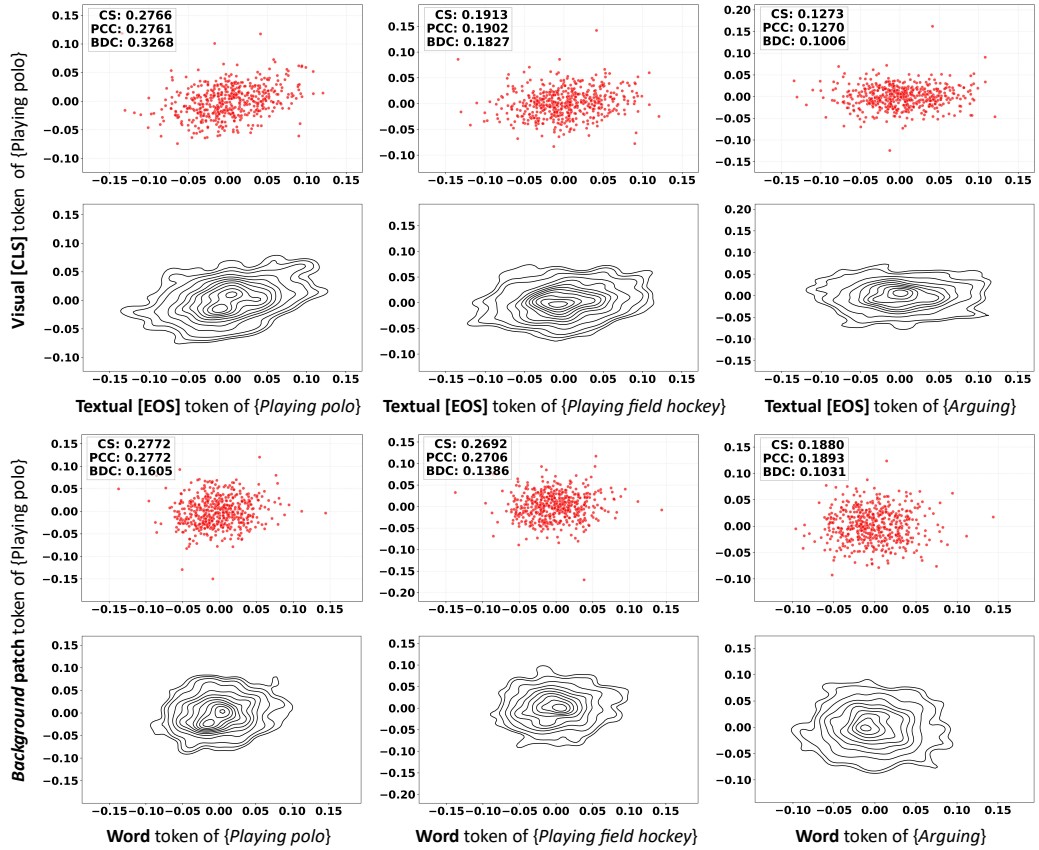

*Figure 4.* Scatterplots and density contours of example visual vs. textual tokens (top: global tokens, bottom: local tokens). The tokens of vision and lanuage exhibit complex relations and their distributions are non-Gaussian. Compared to CS/PCC, BDC effectively distinguishes matching vs. non-matching text-video tokens in the top panel and attenuates irrelevant background tokens in the bottom panel. See Table 6b for dataset-level, apples-to-apples accuracy gains of BDC over CS/PCC.

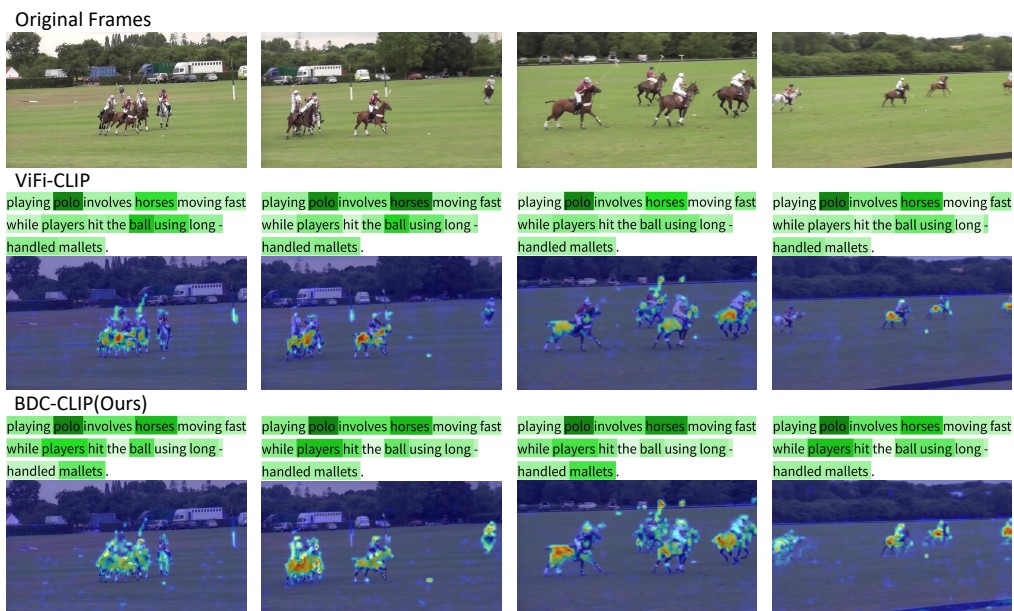

*Figure 5.* Visualizations of {*playing polo*} using Grad-ECLIP. Compared to ViFi-CLIP, our BDC-CLIP better focuses on important image regions such as the horses and players across different frames, while attending more effectively to key words such as 'player' and 'mallets'. *Best viewed by zooming in.*

## Acknowledgments

This work was supported by the National Natural Science Foundation of China (NSFC Nos. 62471083 and 61971086).

## Impact Statement

We propose BDC-CLIP, a novel framework for video-language contrastive learning. Our method can capture complex dependencies between video and text in a shared embedding space. BDC-CLIP holds potential for a range of video understanding tasks such as zero-shot action recognition and video-text retrieval. We hope this work inspires further exploration of BDC and other advanced statistical distances for fine-grained video-language matching.

Since BDC-CLIP is built upon the pre-trained CLIP model, it inevitably inherits some of CLIP's ethical concerns, such as the amplification of pre-existing biases (e.g., gender or racial bias) and the potential for malicious use (Radford et al., 2021). Moreover, as our method relies on LLMs for textual augmentation, it may also be affected by the inherent biases present in these models. We hope acknowledging these potential risks will contribute to development and deployment of more responsible and ethical models.

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

# Appendix

# A. Detailed Experimental Setup

## A.1. Datasets

We conduct experiments on five widely used action recognition datasets, i.e., Kinetics-400 (Carreira & Zisserman, 2017), Kinetics-600 (Carreira et al., 2018), HMDB-51 (Kuehne et al., 2011), UCF-101 (Soomro et al., 2012), and Something-Something v2 (Goyal et al., 2017).

**Kinetics-400 (K400)**  K400 is a large-scale action dataset that spans 400 human action classes each with at least 400 videos collected from YouTube. It provides approximately 240K training videos, 20K validation videos and 40K test videos, capturing a broad spectrum of human actions.

**Kinetics-600 (K600)**  This is an extension of K400 that broadens the scope to 600 categories and $\sim 650$K videos. It has 410K training and 29K validation samples.

**HMDB-51**  This dataset contains 51 action categories and approximately 7K videos manually curated from diverse sources such as movies and YouTube. Three training/validation splits are predefined, each of which contains 3,570 training and 1,530 validation videos.

**UCF-101**  It consists of 101 action classes distributed across over 13K videos sourced primarily from YouTube. It defines three splits for training and validation, each containing roughly 9.5K training and 3.7K validation videos.

**Something-Something v2 (SSv2)**  SSv2 contains more than 100,000 videos across 174 fine-grained action categories, providing about 168K training and 24K validation videos. It emphasizes human-object interactions, and is highly temporal biased compared to other datasets.

## A.2. Task Settings

As in (Rasheed et al., 2023; Kim et al., 2024; Chen et al., 2024b), we benchmark on zero-shot & few-shot recognition, base-to-novel generalization, and fully-supervised recognition. For these tasks, we mainly follow the evaluation protocols of ViFi-CLIP (Rasheed et al., 2023).

**Zero-shot recognition**  We train our models on K400 and benchmark on HMDB-51, UCF-101, and K600. On HMDB-51 and UCF-101, evaluations are conducted on the three official test splits. On K600, as prescribed by Chen & Huang, we use three splits randomly selected from 160 out of the 220 novel categories not present in K400. Following TC-CLIP (Kim et al., 2024) and OST (Chen et al., 2024a), we adopt weight-space ensembling (WSE) technique (Wortsman et al., 2022) for improving zero-shot performance, in which the original CLIP parameters are linearly combined with the fine-tuned parameters with a ratio of 0.3. We report the average accuracy and standard deviation.

**Few-shot recognition**  We evaluate *all-way $K$-shot* tasks on downstream datasets including HMDB-51, UCF-101, and SSv2, where $K = 2, 4, 8, 16$. We randomly sample $K$ videos per category for training, while testing on the first validation split for HMDB-51 and UCF-101, along with the full validation split for SSv2. We experiment in two different few-shot settings: (i) pre-training on K400 followed by fine-tuning on downstream datasets, and (ii) directly fine-tuning from CLIP without pretraining on K400.

**Base-to-novel generalization**  This task involves training models on base (known) classes in few-shot manner while testing on novel (unknown) classes. We evaluate on K400, HMDB-51, UCF-101, and SSv2 using base/novel category splits prescribed by ViFi-CLIP. Three training splits per dataset are constructed. On HMDB-51 and UCF-101, only the first training split is used for training and validation; on K400 and SSv2, evaluations are performed on the full validation set. We also experiment in two different settings: (i) pretraining on K400, and (ii) directly fine-tuining from CLIP without pretraining on K400.

**Fully-supervised recognition**  This is the classical closed-set evaluation. As in ViFi-CLIP, we use standard splits for training and testing. In addition, we employ WSE technique for further performance improvement, where we interpolate ViFi-CLIP zero-shot model with our BDC-CLIP modes with a ratio of 0.2.

## A.3. Hyper-parameter Settings

To achieve better spatio-temporal modeling, we follow the practice of ViCLIP (Wang et al., 2023) by computing attention over the tokens of all frames in a video together at the 4th, 8th, and 12th layers of the CLIP vision encoder.

**Pretraining on K400**  For *zero-shot pretraining*, we use the AdamW optimizer with $\beta_1 = 0.9$, $\beta_2 = 0.98$, a weight decay of 1e-3 and a batchsize of 256. The base learning rate (LR) of the backbone is 8e-6 with a cosine schedule in 10 epochs. The LRs of adapters and vision classifier are $100\times$ and $50\times$ base LR, respectively. We adopt the augmentation techniques and label smoothing as in ViFi-CLIP. For *few-shot pretraining*, the hyper-parameter setting is consistent with zero-shot pretraining, except that the base LR is 4e-6. We sample 32 frames per video and conduct inference with 1 temporal clip and 1 spatial crop (1×1 view).

**Downstream tasks with K400 pretrained models**  The few-shot and base-to-novel settings utilize a batch size of 64, a learning rate of 2e-6 with a cosine schedule in 60 epochs, and a linear warmup over first 5 epochs. We set the learning rate of adapters and vision classifier to $200\times$ and $100\times$ base LR, respectively. The other hyper-parameters align with those used in zero-shot pretraining. We use 32

sampled frames and conduct $1\times 1$ view inference.

**Fully-supervised training on K400** The hyper-parameters in fully-supervised setting are largely consistent with those in the setting of zero-shot pretraining, with a few notable differences. Specifically, the model is trained for 30 epochs that includes 5 linear warmup epochs with a batch size 512; the base LR is set to 2.2e-5 with a cosine schedule. We sample 16 frames per video and conduct inference with 12 views consisting of 4 temporal clips and 3 spatial crops.

# B. Further Experiments

## B.1. Additional Ablation Study about BDC-CLIP

To better understand our BDC-CLIP, we further conduct ablation study on the ratio of tokens used for computing BDC, the impacts of individual visual and textual BDC, and of Token Weighting.

As shown in Table 7, subsampling patch tokens at ratios 3/4, 2/4 and 1/4–or taking only half the word tokens–leads to performance degradation, suggesting retaining all tokens is important for fine-grained alignment. For the impact of applying BDC to the visual and textual branches, we observe that using BDC on only one branch–either vision or text–leads to a clear performance drop, especially on HMDB-51 and SSv2, while applying BDC to both branches achieves optimal results across all benchmarks. Similarly, applying Token Weighting to both branches consistently outperforms using it on a single branch, with weighting only the text branch generally being more effective than weighting only the vision branch.

## B.2. Directly Fine-tuned from CLIP

Here we compare to prior arts in the setting of directly fine-tuned from CLIP, i.e., without pre-training on K400.

**(i) Few-shot recognition** We conduct experiment on HMDB-51, UCF-101 and SSv2 in all-way $K$-shot setting without pretraining on K400. We train BDC-CLIP models with 60 epochs with the base LR of 2e-6 on all three datasets. All methods are directly fine-tuned on the downstream datasets, using $K$ training examples per class for all-way classification. The results, summarized in Table 8, demonstrate that BDC-CLIP achieves superior performance across all three datasets. Specifically, BDC-CLIP consistently surpasses the second-best methods by 2.1%–2.8% on HMDB-51 and 1.0%–1.5% on SSv2 for each of $K$-shot settings. Additionally, BDC-CLIP outperforms the runners-up by 0.6%–1.0% across all settings on UCF-101.

**(ii) Base-to-novel generalization** In this part of experiment, we compare to the competing methods on four datasets in base-to-novel generalization setting. Every method is directly fine-tuned on the base classes and then evaluated on

*Table 7.* Additional Ablation Study about BDC-CLIP.

| Vision | Text | K600 zero-shot | HMDB-51 2-shot | HMDB-51 16-shot | SSv2 2-shot | SSv2 16-shot |
|---|---|---|---|---|---|---|
| Effect of Token Ratio | | | | | | |
| 4/4 | 4/4 | $73.8_{\pm 0.8}$ | 66.1 | 73.9 | 8.9 | 16.8 |
| 3/4 | 4/4 | $73.7_{\pm 0.7}$ | 64.5 | 73.7 | 8.6 | 16.5 |
| 2/4 | 4/4 | $73.4_{\pm 0.7}$ | 65.8 | 74.9 | 8.3 | 16.5 |
| 1/4 | 4/4 | $73.6_{\pm 0.7}$ | 64.7 | 73.8 | 8.4 | 16.4 |
| 4/4 | 2/4 | $73.4_{\pm 0.7}$ | 65.5 | 73.2 | 8.6 | 16.4 |
| Visual and/or Textual BDC | | | | | | |
| ✓ | ✗ | $73.6_{\pm 0.8}$ | 65.2 | 72.3 | 7.9 | 16.0 |
| ✗ | ✓ | $73.8_{\pm 0.8}$ | 64.3 | 71.5 | 7.5 | 14.3 |
| ✓ | ✓ | $73.8_{\pm 0.8}$ | 66.1 | 73.9 | 8.9 | 16.8 |
| Effect of Token Weighting | | | | | | |
| ✓ | ✗ | $73.2_{\pm 1.1}$ | 65.7 | 73.2 | 7.6 | 16.2 |
| ✗ | ✓ | $74.0_{\pm 0.7}$ | 66.1 | 73.4 | 8.1 | 16.1 |
| ✓ | ✓ | $73.8_{\pm 0.8}$ | 66.1 | 73.9 | 8.9 | 16.8 |

novel classes in zero-shot manner. Our BDC-CLIP models are trained in 12 epochs, with the base LRs of 2e-6 on HMDB-51 & UCF-101, 4e-6 on K400 and 5e-6 on SSv2. Table 9 reports the Top-1 accuracy for the base and novel classes, along with their harmonic mean (HM). The results indicate that BDC-CLIP achieves state-of-the-art performance across all three datasets. Specifically, in terms of HM, BDC-CLIP surpasses the previous best-performing method TC-CLIP by 0.5% on K400, and ~1.0% on all other three datasets.

It is worth noting that most methods in this directly fine-tuned setting perform remarkably inferior to their individual counterparts in the setting of pretraining on K400, for either few-shot recognition (Table 2 vs. Table 8) or base-to-novel generalization (Table 3 vs. Table 9). This comparison suggests that pretraining on large-scale K400 is pivotal in bridging the image and video gaps for CLIP, significantly benefiting downstream video action recognition tasks.

## B.3. BDC-CLIP for Few-shot IMAGE Recognition

Our BDC-CLIP can be extended to few-shot image recognition. To adapt to image data, we remove the temporal attention module from the visual encoder. Given the limited number of training images, we adopt a parameter-efficient approach similar to CLIP-LoRA (Zanella & Ben Ayed, 2024) by attaching a LoRA (Hu et al., 2022) module (rank 2, alpha 1) to each transformer block in both textual and visual encoders. We use CLIP model with ViT-B/16 as image encoder, and conduct experiments under the 16-shot setting. For the training hyperparameters, we follow the same settings as CLIP-LoRA. Following standard practice, we evaluate our approach on ImageNet (Deng et al., 2009), Aircraft (Maji et al., 2013), Food (Bossard et al., 2014), DTD (Cimpoi et al., 2014), UCF101 (Soomro et al., 2012), Cars (Krause et al., 2013), OxfordPets (Parkhi et al.,

| Method | BEs | HMDB-51 | | | | UCF-101 | | | | SSv2 | | | |
|---|---|---|---|---|---|---|---|---|---|---|---|---|---|
| | | K=2 | K=4 | K=8 | K=16 | K=2 | K=4 | K=8 | K=16 | K=2 | K=4 | K=8 | K=16 |
| CLIP [Radford et al.] | ❄ | 41.9 | 41.9 | 41.9 | 41.9 | 63.6 | 63.6 | 63.6 | 63.6 | 2.7 | 2.7 | 2.7 | 2.7 |
| A5 [Ju et al.] | ❄ | 39.7 | 50.7 | 56.0 | 62.4 | 71.4 | 79.9 | 85.7 | 89.9 | 4.4 | 5.1 | 6.1 | 9.7 |
| ActionCLIP [Wang et al.] | 🔥 | 47.5 | 57.9 | 57.3 | 59.1 | 70.6 | 71.5 | 73.0 | 91.4 | 4.1 | 5.8 | 8.4 | 11.1 |
| X-CLIP [Ni et al.] | 🔥 | 53.0 | 57.3 | 62.8 | 64.0 | 76.4 | 83.4 | 88.3 | 91.4 | 3.9 | 4.5 | 6.8 | 10.0 |
| ViFi-CLIP [Rasheed et al.] | 🔥 | 57.2 | 62.7 | 64.5 | 66.8 | 80.7 | 85.1 | 90.0 | 92.7 | 6.2 | 7.4 | 8.5 | 12.4 |
| OST [Chen et al.] | 🔥 | 59.1 | 62.9 | 64.9 | 68.2 | 82.5 | 87.5 | 91.7 | 93.9 | 7.0 | 7.7 | 8.9 | 12.2 |
| TC-CLIP [Kim et al.] | 🔥 | 58.6 | 63.3 | 65.5 | 68.8 | 86.8 | 90.1 | 92.0 | 94.3 | 7.3 | 8.6 | 9.3 | 14.0 |
| BDC-CLIP (Ours) | 🔥 | 61.4 | 65.6 | 67.9 | 70.9 | 87.4 | 90.8 | 93.0 | 95.1 | 8.3 | 9.6 | 10.6 | 15.5 |

*Table 8.* Results of *few-shot* VIDEO *recognition* without pretraining on K400. All methods are directly fine-tuned from CLIP in $K$-shot settings. CLIP Backbone Encoders (BEs) are frozen (❄) or fine-tuned (🔥). The best results are **bold** and the second-best ones are underlined.

| Method | BEs | K400 | | | HMDB-51 | | | UCF-101 | | | SSv2 | | |
|---|---|---|---|---|---|---|---|---|---|---|---|---|---|
| | | Base | Novel | HM | Base | Novel | HM | Base | Novel | HM | Base | Novel | HM |
| CLIP [Radford et al.] | ❄ | 62.3 | 53.4 | 57.5 | 53.3 | 46.8 | 49.8 | 78.5 | 63.6 | 70.3 | 4.9 | 5.3 | 5.1 |
| A5 [Ju et al.] | ❄ | 69.7 | 37.6 | 48.8 | 46.2 | 16.0 | 23.8 | 90.5 | 40.4 | 55.8 | 8.3 | 5.3 | 6.4 |
| ActionCLIP [Wang et al.] | 🔥 | 61.0 | 46.2 | 52.6 | 69.1 | 37.3 | 48.5 | 90.1 | 58.1 | 70.7 | 13.3 | 10.1 | 11.5 |
| X-CLIP [Ni et al.] | 🔥 | 74.1 | 56.4 | 64.0 | 69.4 | 45.5 | 55.0 | 89.9 | 58.9 | 71.2 | 8.5 | 6.6 | 7.4 |
| ViFi-CLIP [Rasheed et al.] | 🔥 | 76.4 | 61.1 | 67.9 | 73.8 | 53.3 | 61.9 | 92.9 | 67.7 | 78.3 | 16.2 | 12.1 | 13.9 |
| Open-VCLIP [Weng et al.] | 🔥 | 76.5 | 62.6 | 68.9 | 70.3 | 50.4 | 58.7 | 94.8 | 77.5 | 85.3 | 16.0 | 11.0 | 13.0 |
| FROSTER [Huang et al.] | 🔥 | 77.8 | 64.3 | 70.4 | 74.1 | 58.0 | 65.1 | 95.3 | 80.0 | 87.0 | 18.3 | 12.2 | 14.6 |
| TC-CLIP [Kim et al.] | 🔥 | 79.1 | 65.4 | 71.6 | 73.3 | 59.1 | 65.5 | 95.4 | 81.6 | 88.0 | 17.5 | 13.4 | 15.2 |
| BDC-CLIP (Ours) | 🔥 | 79.9 | 65.6 | 72.1 | 75.0 | 59.5 | 66.4 | 95.6 | 83.7 | 89.1 | 18.8 | 14.9 | 16.6 |

*Table 9.* Results of *base-to-novel* VIDEO *recognition* without pretraining on K400. All methods are directly fine-tuned from CLIP on base classes, and then evaluated on validation set of base classes along with on novel classes in zero-shot manner.

*Table 10.* Results of *few-shot* IMAGE *recognition*. All methods are compared using CLIP models with ViT-B/16 as image encoder in the 16-shot setting. CLIP Backbone Encoders (BEs) are frozen (❄) or finetuned (🔥).

| Method | BEs | ImageNet | Aircraft | Food101 | DTD | UCF101 | Cars | Pets | SUN397 | Flowers102 | Caltech101 | EuroSAT | Average |
|---|---|---|---|---|---|---|---|---|---|---|---|---|---|
| BDC-Adapter [Zhang et al.] | ❄ | 66.5 | 39.5 | 80.5 | 71.1 | 86.5 | 78.8 | 92.0 | 72.7 | 97.0 | 93.9 | 85.2 | 78.5 |
| TransCLIP [Zanella et al.] | ❄ | 71.8 | 38.6 | 86.9 | 65.1 | 82.1 | 79.8 | 92.4 | 74.7 | 94.4 | 94.0 | 83.0 | 78.4 |
| ProGrad [Zhu et al.] | ❄ | 72.1 | 43.0 | 85.8 | 68.8 | 82.7 | 71.9 | 36.8 | 75.1 | 96.6 | 95.9 | 83.6 | 79.9 |
| LLaMP [Zheng et al.] | ❄ | 73.5 | 56.1 | 87.6 | 74.2 | 86.8 | 86.1 | 94.2 | 77.0 | 98.1 | 97.1 | 91.3 | 83.8 |
| CLIP-LoRA [Zanella & Ben Ayed] | 🔥 | 73.6 | 54.7 | 84.2 | 72.0 | 86.7 | 86.3 | 92.3 | 76.1 | 98.0 | 96.4 | 92.1 | 83.0 |
| BDC-CLIP (Ours) | 🔥 | 75.0 | 57.3 | 88.1 | 76.5 | 87.7 | 86.5 | 94.4 | 78.3 | 98.4 | 97.3 | 93.9 | 84.9 |

2012), SUN397 (Xiao et al., 2010), Flowers102 (Nilsback & Zisserman, 2008), Caltech101 (Fei-Fei et al., 2004), EuroSAT (Helber et al., 2019).

As shown in Table 10, BDC-CLIP surpasses CLIP-LoRA (Zanella & Ben Ayed, 2024), a strong baseline, by 1.9%, and outperforms the second-best method, i.e., LLaMP (Zheng et al., 2024), by 1.1%. Notably, BDC-CLIP achieves the highest performance across all 11 datasets. The comparison suggests that BDC-CLIP that uses BDC and all visual and textual tokens is effective for both video and image recognition tasks.

### B.4. Extra Visualizations

**t-SNE visualization of video-language representation** We compute prediction probability vectors with K400-pretrained models for video samples from the validation sets of HMDB-51 and K600. To isolate the influence of LLMs, we use the CLIP prompt template for zero-shot classification. These probability vectors are projected into a 2D space using t-SNE. Figure 6 presents the results, with different categories represented by distinct colors.

On HMDB-51 (Figure 6a), BDC-CLIP exhibits better inter-class separability compared to the strong baseline of ViFi-CLIP, and significantly outperforms vanilla CLIP. Similar trends are observed on K600 (Figure 6b). We further evaluate clustering quality using Adjusted Rand Index (ARI) and Normalized Mutual Information (NMI), where BDC-CLIP achieves higher scores for both metrics, consistent with the visual results. These comparisons suggest that the BDC metric, leveraging all visual and textual tokens, can learn representations with stronger generalization capabilities compared to the cosine similarity relying only on global tokens.

**Visualization of video-text pairs** We compare models pretrained on K400 and evaluated in a zero-shot setting on representative clips from K600. Figure 7 shows that BDC–CLIP consistently highlights salient image regions and words more accurately than ViFi-CLIP. For {*Bulldozing*}, ViFi-CLIP is distracted by most of irrelevant regions (i.e., ground) while BDC centers on the machine and material without distraction; meanwhile, BDC-CLIP better attends to key words such as 'scraping'. For {*Bull fighting*}, BDC-

CLIP better focuses on the image regions of the bull, mata-dor, and cape across the frames; in the meantime, BDC-CLIP pays more attention to key words such as 'bull fight-ing' and 'rushes' in the textual description. These examples demonstrate BDC-CLIP's ability to capture rich spatial, temporal, and linguistic context. We attribute this to the BDC metric and the inclusion of all local tokens, which to-gether enable the modeling of complex, fine-grained video-language relations.

## C. Limitations and Future Work

BDC-CLIP exhibits somewhat higher GFLOPs and lower throughput compared to state-of-the-art methods. However, as shown in Table 5, this overhead remains moderate. While BDC effectively captures nonlinear correlations, other met-rics, such as mutual information (MI), offer similar capabil-ities. However, MI faces computational challenges in high-dimensional spaces due to density estimation requirements and the difficulty of lower-bound estimation (Belghazi et al., 2018). Exploring such alternatives presents an exciting av-enue for future work. Currently, BDC-CLIP is tailored for video action recognition. Extending it to other tasks, such as video-text retrieval or open-world object detection in videos, represents a promising direction for future research.

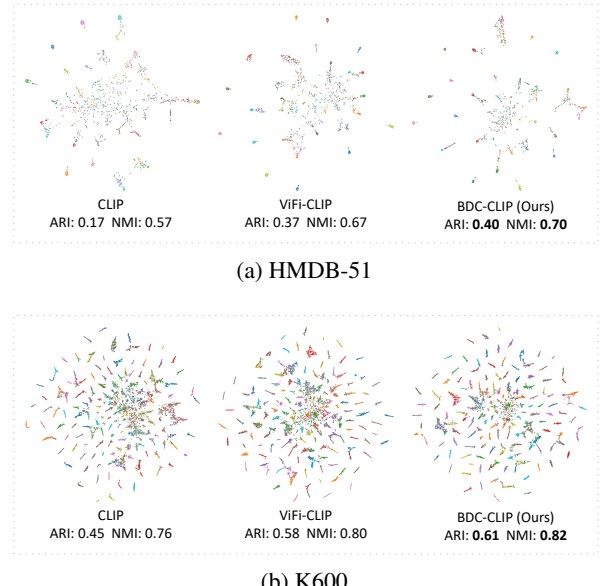

(a) HMDB-51

(b) K600

*Figure 6.* t-SNE visualizations of joint video-language representa-tions. K400-pretrained models are used for zero-shot recognition on *HMDB-51* and *K600*. BDC-CLIP shows better separability than ViFi-CLIP and vanilla CLIP; the quantitative results of the two clustering quality measures (i.e., ARI and NMI) are consistent with the visualization results. *Best viewed by zooming in.*

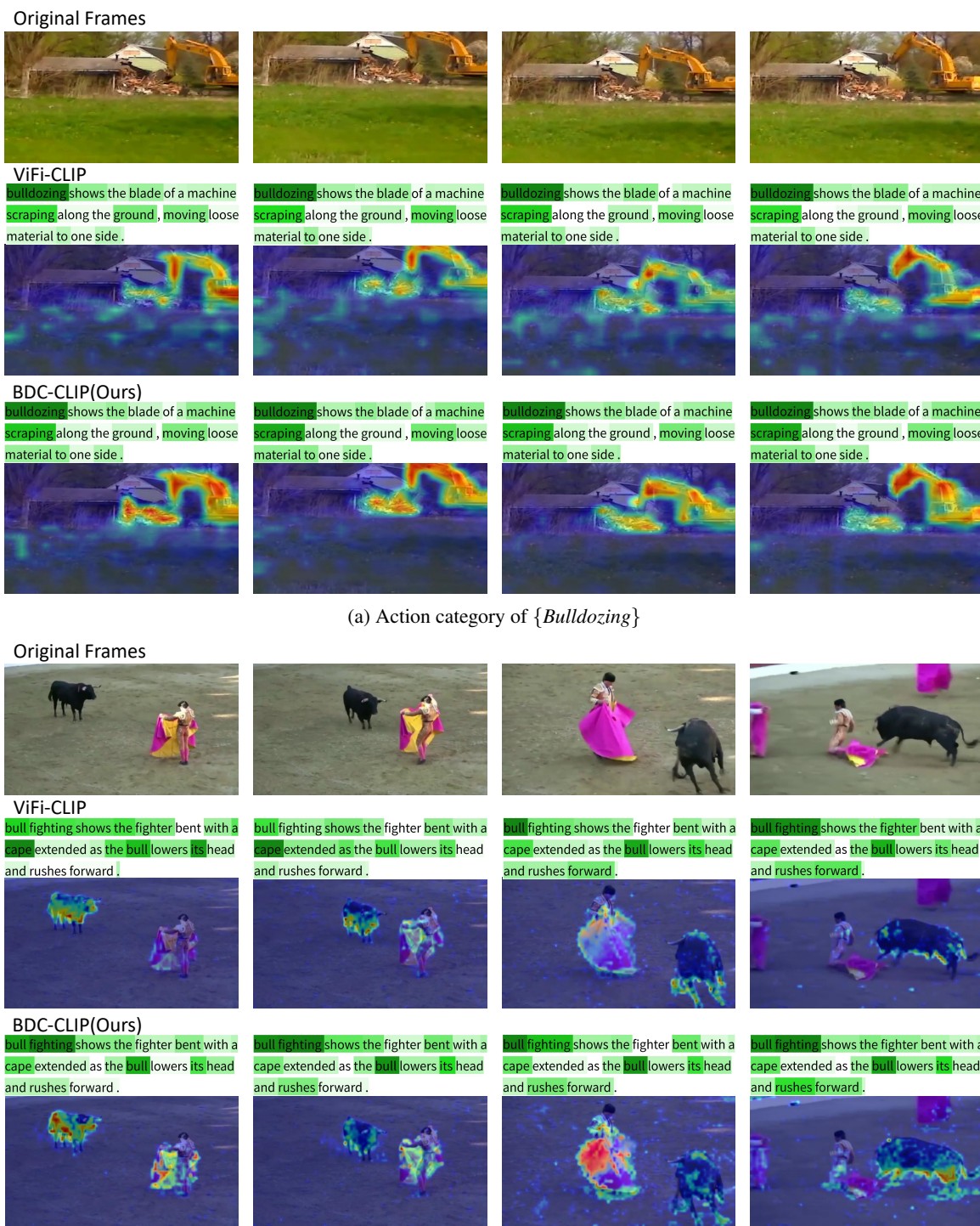

(a) Action category of {*Bulldozing*}

(b) Action category of {*Bull fighting*}

*Figure 7.* Heat-maps for representative video-text pairs produced with Grad-ECLIP using the K400-pre-trained BDC-CLIP in zero-shot setting. Compared with ViFi-CLIP, BDC-CLIP more sharply highlights both the action-critical image regions and the corresponding salient words as they evolve over time, demonstrating its ability to capture fine-grained multimodal context across space, time, and language. *Best viewed by zooming in.*

