# OpenReview forum: "BDC-CLIP: Brownian Distance Covariance for Adapting CLIP to Action Recognition"
_ICML.cc/2025/Conference — ICML 2025 poster_

### Official Review · Reviewer_sjZc · 2025-02-19

**Overall Recommendation:** 3

**Summary:**

In this work, the authors propose an adapter-based action recognition method built on top of CLIP visual and text encoders. To better capture local details, they propose to utilize all visual patch tokens and word tokens and employ Brownian Distance Covariance (BDC) as a similarity metric between video and text representations. They validate the proposed method on several public datasets and tasks.


## update after rebuttal
The additional evidence is somewhat weak as it is based on only a couple of samples or one action class. Still, ~20% accuracy on SSv2 is far from being useful, similar to other CLIP-based methods. So I would say this is still a downside of this work. Therefore, I would like to keep my original rating.

**Claims And Evidence:**

Some of the claims made are not fully supported by evidence.
- “BDC captures both linear and nonlinear relations, enabling it to model the complex dependencies in the video-language embedding space” -> There is no evidence of BDC enabling complex dependency modeling in the video-language embedding space. The paper only shows favorable task performance, some feature space visualization and attention map visualization.
- “…enabling fine-grained multi-modal context modeling in space, time, and language” -> Similar to the first claim above, there is no evidence of *fine-grained multi-modal context modeling* in space, time, and language in the paper.

**Essential References Not Discussed:**

Please consider citing the following works and discussing the relation to the proposed method.

- [Park et al., Dual-path Adaptation from Image to Video Transformers, CVPR 2023]
- [Lee et al., CAST: Cross-Attention in Space and Time for Video Action Recognition, NeurIPS 2023]
- [Qian et al., Rethinking Image-to-Video Adaptation: An Object-Centric Perspective, ECCV 2024]

**Experimental Designs Or Analyses:**

The proposed method needs to be further validated by more experiments.
- What happens if we do not employ temporal attention in eq (4)? (effect of the proposed temporal modeling method)
- What happens if we use 3/4, 2/4, 1/4 of the patch tokens or half of the word tokens instead of using them all?
- What happens if we turn on and off (or replace it with cosine similarity) text BDC matrices computing and visual BDC matrices computing? Which BDC computing is more important?
- What happens if we turn on and off text token weighting and visual token weighting?

**Methods And Evaluation Criteria:**

The proposed method is sensible as we can reuse a vision-language pre-trained model with relatively little amount of fine-tuning effort. Employing BDC as similarity metric to model non-linear relation between tokens make sense and is proven effective in action recognition to some extent. Downside of the method is the limited capability of temporal dynamics modeling, similar to prior CLIP-based works such as Action-CLIP, X-CLIP, TC-CLIP, etc, as shown in SSV2 experiments in Table 2 and 3.

**Other Comments Or Suggestions:**

- L195: “Let the values be V=[…” -> where does V come from?
- L211-213: “we first achieve the embeddings of reduced dimension …” -> awkward sentence
- L417: “... models are valuated ...” -> evaluated?

**Other Strengths And Weaknesses:**

Please address my concerns on the unsupported claims, temporal modeling capability of the method, missing empirical validation.

**Questions For Authors:**

Please address my concerns on the unsupported claims, temporal modeling capability of the method, missing empirical validation.
1) I do not understand how to construct $\textbf{b}^t=Vech(\textbf{B}^t) \in \mathbb{R}^{d(d+1)/2}$ in L190-192. Can the authors elaborate on this?

**Relation To Broader Scientific Literature:**

Since this work is built on top of CLIP [Radford et al., ICML 2021], it is related to the vision-language alignment and multi-modal learning field. Similar to prior works using CLIP, the proposed method also shows limitations in modeling temporal dynamics as demonstrated in the Something-Something-V2 experiments in Table 2 and 3. All the CLIP-family including the proposed method show unfavorable performance on SSV2: ~20% accuracy while SOTA method with similar backbone (ViT-B) shows >70% accuracy on this dataset.

**Theoretical Claims:**

There is no theoretical claims made.

---

> ### Author Rebuttal · Authors · 2025-04-01
>
> Dear Reviewer sjZc,
>
> We sincerely thank you for your constructive and insightful comments, particularly your positive feedback & decision.
>
> > ### Q1: The paper lacks evidence demonstrating that BDC enables modeling of complex dependencies in the video-language embedding space.
>
> Thanks for this concern.
>
> In BDC-CLIP, we compare two sets of $d$-dimensional tokens from language and vision. According to Zhelezniak et al. (2019), this setup can be viewed as modeling dependencies between $d$-observation samples from two random variable sets (textual vs. visual tokens). Furthermore, **Szekely & Rizzo (2009) show that BDC captures all types of statistical dependencies--linear and nonlinear--without assumptions on joint distributions.** Hence, by applying BDC to video-language alignment, BDC-CLIP can effectively model complex dependencies in the shared embedding space. *The scatterplots and token distributions (kindly see [Figure R1](https://anonymous.4open.science/r/rebuttal-D15F/Figure_R1.jpg)) illustrate that our model can effectively learn nonlinear relationships.*
>
>
> >  ### Q2: The paper does not provide evidence to support the claimed fine-grained multi-modal context modeling across spatial, temporal, and language dimensions.
>
> The set of textual tokens naturally encodes key nouns and verbs describing people and objects, while the set of patch tokens captures crucial spatial regions in each video frame. As BDC can measure any form of statistical dependency between textual and visual tokens, **it provides a principled way to capture rich relations across spatial and language dimensions.** Further, by averaging BDC matrices over consecutive frames, our method models temporal evolution of the token-level correspondences.
>
> The heatmaps (kindly see [Figure R2](https://anonymous.4open.science/r/rebuttal-D15F/Figure_R2.jpg)) show how the model focuses on horses and players in the {playing polo}--highlighting the relevant objects and interactions in both text and video frames. **These results suggest that BDC-CLIP indeed learns fine-grained, action-centric context spanning space, time, and language.**
>
>
> >  ### Q3: BDC-CLIP needs to be further validated by more experiments: 1) without (w/o) temporal attention (TA), 2) using 3/4, 2/4, 1/4 of patch tokens (PT) or half of the word tokens (WT), 3) turn off text or visual BDC matrices, and 4) turn off text and visual tokens weighting (TW).
>
> Thank you for the suggestions.
>
>  1)$\ $W/O TA, BDC-CLIP’s performance drops consistently across 3 datasets, indicating TA provides valuable temporal modeling. 2) Subsampling patch tokens at ratios 3/4 and 1/4--or taking only half the word tokens--also degrades results, suggesting retaining all tokens is important for fine-grained alignment. 3) Removing either text or visual BDC significantly hurts performance, with a more pronounced drop from removing visual BDC, suggesting more importance of the visual adapter. 4) Disabling TW for either text or vision negatively impacts performance, underscoring how focusing on more informative tokens benefits our approach.
>
>
> |Method|K600_Zeroshot|HMDB51_2shot|HMDB51_16shot| SSv2_2shot|SSv2_16shot|
> |:-:|:-:|:-:|:-:|:-:|:-:|
> |BDC-CLIP|73.8$\pm$0.8|66.1|73.9|8.9|16.8|
> | w/o TA|73.3$\pm$0.8|65.5|73.4|8.3|16.2|
> |3/4 PTs|73.7$\pm$0.8|64.5|73.7|8.6|16.5|
> |1/4 PTs|73.7$\pm$0.9|64.7|73.8|8.5|16.5|
> |1/2 WTs|73.4$\pm$0.8|65.5|73.2|8.6|16.4|
> |w/o text BDC|73.6$\pm$0.8|65.2|72.3|7.9|16.0|
> |w/o visual BDC|73.8$\pm$0.8|64.3|71.5|7.5|14.3|
> |w/o text TW|73.2$\pm$1.1|65.7|73.2|7.6|16.2|
> |w/o visual TW|74.0$\pm$0.7|66.1|73.4|8.1|16.1|
>
> >  ### Q4: Similar to prior works using CLIP, BDC-CLIP shows limitations in modeling temporal dynamics as demonstrated on SSv2 in Table 2 and 3, achieving ~20% accuracy while SOTA method with similar backbone (ViT-B) shows >70% accuracy.
>
> Kindly note our reported ∼20% or lower accuracy in Tables 2 and 3 are from few-shot and base-to-novel settings, while the mentioned SOTA results (>70%) rely on fully supervised (FullS) training. **These differing settings are not directly comparable**, and it remains unclear how those FullS approaches would fare in few-shot or base-to-novel settings.
>
>
>
> > ### Q5: Essential References Not Discussed.
>
> Thanks for highlighting the three works that will be cited and discussed in our revision. Briefly, **they focus on adapting pure vision-pretrained models for fully-supervised (FullS) action recognition**. In contrast, **BDC-CLIP relies on multimodal alignment between vision and language,** achieving strong performance in, beside FullS setting, zero-shot, few-shot and base-to-novel settings.
>
> >  ### Q6: How to construct $\mathbf{b_t}$ in L190-192?
>
> $\mathbf{B_t}$ is a symmetric $d\times d$ matrix, so half-vectorization (Vech) collects the elements on or below its diagonal into a $d(d+1)/2$-dimensional vector $\mathbf{b_t}$. Kindly see the [Wikipedia article](https://en.wikipedia.org/wiki/Vectorization_(mathematics)#Half-vectorization) for details on Vech.

---

> > ### Comment · Reviewer_sjZc · 2025-04-09
> >
> > Thanks for the rebuttal response. I have read the rebuttal and other reviews. The rebuttal partially resolved my concerns. I appreciate partial evidence on BDC enabling modeling of complex dependencies and fine-grained multi-modal context modeling capability in https://anonymous.4open.science/r/rebuttal-D15F/Figure_R1.jpg and https://anonymous.4open.science/r/rebuttal-D15F/Figure_R2.jpg. However, the evidence is somewhat weak as it is based on only a couple of samples or one action class. I understand that Table 2 and 3 are few-shot and base-to-novel results. Still, ~20% accuracy on SSv2 is far from being useful, similar to other CLIP-based methods. So I would say this is still a downside of this work. Therefore, I would like to keep my original rating.

---

> > > ### Author Response · Authors · 2025-04-09
> > >
> > > Dear Reviewer sjZc,
> > >
> > > Thank you for your thoughtful feedback. We are glad our rebuttal has partially resolved your concerns, and we appreciate this opportunity to clarify the remaining points.
> > >
> > > > ### Q1': I appreciate partial evidence on BDC enabling modeling of complex dependencies and fine-grained multi-modal context modeling capability in [Figure R1](https://anonymous.4open.science/r/rebuttal-D15F/Figure_R1.jpg) and [Figure R2](https://anonymous.4open.science/r/rebuttal-D15F/Figure_R2.jpg). The additional evidence is somewhat weak as it is based on only a couple of samples or one action class.
> > >
> > > Our vision–language matching framework uses **BDC  (Szekely & Rizzo, 2009)**, a robust statistical metric capable of capturing *any* form of dependency between random variables. By measuring similarity among all visual tokens (fine-grained spatial regions) and all textual tokens (linguistic elements) across frames, our approach models rich multimodal context spanning space, language, and time.
> > >
> > > We believe **it is BDC's strong theoretical properties** that enable modeling of complex dependencies in the shared embedding space, thereby enabling fine-grained multimodal alignment. These capabilities are **further supported by our detailed ablation studies and broad experiments** (zero-shot, few-shot, base-to-novel, fully supervised) *presented in the main paper.*
> > >
> > > **Our additional evidence—scatterplots, token distributions, and attention heatmaps—serves as an *intuitive illustration* of these strengths**. We apologize for limiting these examples to just a couple of samples or one action class, *due to the rebuttal’s time constraints.* **We plan to add more examples in the revised version** to further highlight our method’s modeling ability.
> > >
> > > > ### Q2': Still, ~20% accuracy on SSv2 is far from being useful, similar to other CLIP-based methods. So I would say this is still a downside of this work.
> > >
> > > We understand your concerns about SSv2 performance and *acknowledge this limitation.* However, **we believe these results should not be viewed as a fundamental drawback of our proposed method.** Rather, they reflect the **extreme difficulty of few-shot (*all-way K-shot*) and base-to-novel recognition on SSv2**—scenarios that challenge a wide range of approaches, including other CLIP-based methods (e.g., Action-CLIP, X-CLIP, TC-CLIP).
> > >
> > > We see these lower numbers not as a reason to dismiss CLIP-based approaches, but as a call to explore new ideas and strategies that can ultimately lead to *practical* success in demanding scenarios. We hope our **BDC-CLIP will serve as a stepping stone** for developing more robust techniques on SSv2 and similarly challenging tasks.
> > >
> > > Once again, we thank you for your time and feedback, and we trust this additional response clarifies your remaining concerns.
> > >
> > > Sincerely,
> > >
> > > The Authors

---

### Official Review · Reviewer_LE28 · 2025-02-24

**Overall Recommendation:** 4

**Summary:**

This paper proposes BDC-CLIP, a framework that introduces Brownian Distance Covariance (BDC) to address the limitations of current CLIP-based video models. BDC-CLIP can leverage all the visual and textual embeddings and construct non-linear relations for vision-language modeling. BDC-CLIP achieves state-of-the-art performance on multiple video benchmarks under various settings.

## update after rebuttal
Thanks for your rebuttal. The concerns have been solved. I will increase the score to 4.

**Claims And Evidence:**

The claims are well-supported in the paper.

**Essential References Not Discussed:**

N/A

**Experimental Designs Or Analyses:**

Yes, in Sec. 4.

**Methods And Evaluation Criteria:**

Yes.

**Other Comments Or Suggestions:**

N/A

**Other Strengths And Weaknesses:**

### Strength

- The motivation is clear and intuitive, which aims to address an intrinsic limitation for current video models.

- The achieved performance is great on a range of benchmarks and experimental settings.

### Weaknesses

- The core technique, i.e., Brownian Distance Covariance, is proposed by a previous paper. Therefore, the technical contribution of this paper seems a bit limited.

- Additional ablation experiments should be added. In the paper, I cannot see how the performance gradually improves on top of a vanilla baseline with all the proposed modules. The authors should elaborate more on this.

- What features will the model use for inference: global features, local features or BDC matrices?

- The proposed method should be compatible with image-only tasks, e.g., few-shot image classification. I suggest the authors conducting some experiments on that for further demonstrating the effectiveness of the paper.

**Questions For Authors:**

N/A

**Relation To Broader Scientific Literature:**

The proposed method can achieve state-of-the-art performance on a wide range of video benchmarks.

**Theoretical Claims:**

N/A

---

> ### Author Rebuttal · Authors · 2025-04-01
>
> Dear Reviewer LE28,
>
> We sincerely thank you for providing constructive and insightful comments. In particular, we appreciate your positive comments including  **"The motivation is clear and intuitive"** as well as **"The achieved performance is great on a range of benchmarks and experimental settings."**
>
>
> > ### Q1: The technical contribution appears somewhat limited, as the core technique, Brownian Distance Covariance, was already introduced in prior work.
>
> Thanks for your concern.
>
> To our best knowledge, our work is the first attempt that introduces BDC into both text and visual encoders of CLIP for vision-language alignment. **Our work highlights, *in foundation models such as CLIP,* potential of advanced metric (e.g., BDC) over ubiquitous cosine similarity.** In contrast, DeepBDC (Xie et al., 2022) concerns no CLIP framework, while BDC-adapter (Zhang et al., 2023) only uses BDC in single vision modality. Besides, we design **temporal attention mechanism for  BDC representations that previous arts lack** as they focus on image recognition. *kindly see Lines 151-164 in the paper for detailed discussion on differences from the previous works.*
>
> In terms of the above clarifications, we would be grateful if you could re-evaluate our technical contributions.
>
>
> > ### Q2: Additional ablation experiments should be added. In the paper, I cannot see how the performance gradually improves on top of a vanilla baseline with all the proposed modules. The authors should elaborate more on this.
>
> Thank you for the comment.
>
> Kindly note that we present the **performance variation on top of a vanilla baseline *in Table 6a.*** Following your suggestion, we **add additional ablation experiments** for further illustrating the effect of proposed modules. Specifically, we evaluate BDC-CLIP for the following settings: 1) not employ temporal attention (TA), 2) using 3/4, 1/4 of patch tokens or half of the word tokens, 3) turn off text or visual BDC matrices, and 4) turn off text and visual tokens weighting (TW).  ***Kindly refer to our response to Q3 of Reviewer sjZc for the results and discussion.***
>
>
>
> > ### Q3: What features will the model use for inference: global features, local features or BDC matrices?
>
> Our inference relies primarily on the BDC matrices produced by the two adapters for both video-language classification and purely visual classification. In addition, as in prior works, we also incorporate global features from the backbone encoder to enable a standard CLIP-like classification branch.
>
> > ### Q4: The proposed method should be compatible with image-only tasks, e.g., few-shot image classification. I suggest the authors conducting some experiments on that for further demonstrating the effectiveness of the paper.
>
> Thanks for your comment.
>
> **As suggested, we extend BDC-CLIP to few-shot image recognition.** Specifically, we remove temporal attention module in the visual encoder to fit for image recognition task; as the training images are scarce, we adopt parameter efficient technique as in CLIP-LoRA, attaching a LoRA module (rank 2, alpha 1) to each transformer block for both textual and visual encoders. Following previous arts, we conduct comparison on 11 datasets with ViT-B/16 as the visual encoder in 16-shot setting. From the table below, we see our BDC-CLIP improves over the strong baseline of CLIP-LoRA by 1.9\%, while outperforming the second-best (i.e, LLaMP) by 1.1\%. Notably, BDC-CLIP stands out across all 11 datasets. **The comparison suggests that BDC-CLIP that uses BDC and all patch tokens is general, effective for both video and image recognition tasks.**
>
> |Method|ImageNet|Aircraft|Food| DTD|UCF|Cars|Pets|SUN|Flowers|Caltech|EuroSAT|Avg|
> |:-|:-:|:-:|:-:|:-:|:-:|:-:|:-:|:-:|:-:|:-:|:-:|:-:|
> |TransCLIP (Zanella et al.)|71.8|38.6|86.9|65.1|82.1|79.8|92.4|74.7|94.4|94.0|83.0|78.4|
> |ProGrad (Zhu et al.)|72.1|43.0|85.8|68.8|82.7|71.9|36.8|75.1|96.6|95.9|83.6|79.9|
> |CLIP-LoRA (Maxime et al.) |73.6|54.7|84.2|72.0|86.7|86.3|92.3|76.1|98.0|96.4|92.1|83.0|
> |LLaMP (Zheng et al.)|73.5|56.1|87.6|74.2|86.8|86.1|94.2|77.0|98.1|97.1|91.3|83.8|
> |BDC-CLIP (ours) |75.0|57.3|88.1|76.5|87.7|86.5|94.4|78.3|98.4|97.3|93.9|84.9|
>
> * Zanella M, Gerin B, Ayed I. Boosting vision-language models with transduction. In NeurIPS, 2024.
> * Zhu B, Niu Y, Han Y, et al. Prompt-aligned gradient for prompt tuning. In CVPR, 2023.
> * Maxime Z, Ismail B A. Low-rank few-shot adaptation of vision language models. In CVPRW, 2024.
> * Zheng Z, Wei J, Hu X, et al. Large language models are good prompt learners for low-shot image classification. In CVPR, 2024.

---

> > ### Comment · Reviewer_LE28 · 2025-04-02
> >
> > Thanks for your rebuttal. The concerns have been solved. I will increase the score to 4.

---

> > > ### Author Response · Authors · 2025-04-02
> > >
> > > Dear Reviewer LE28,
> > >
> > > We appreciate that our rebuttal has addressed your concerns, and we are grateful for your decision to raise your score. Thank you for supporting our work.
> > >
> > > Sincerely,
> > >
> > > The Authors

---

### Official Review · Reviewer_oB1H · 2025-03-12

**Overall Recommendation:** 3

**Summary:**

This paper proposes BDC-CLIP, a novel framework for video-language alignment based on Brownian Distance Covariance (BDC). Unlike cosine similarity, BDC can capture both linear and nonlinear correlations. BDC-CLIP leverages all visual and textual tokens to model both linear and nonlinear relationships in the multimodal embedding space, thereby capturing rich contextual information across spatial, temporal, and linguistic dimensions. The framework also introduces a temporal BDC attention mechanism that integrates patch-wise spatial cues and frame-wise temporal dynamics.

## update after rebuttal
I tend to give a borderline score (between 2 and 3), but since ICML only has a weak accept option, I have raised the score to 3.

**Claims And Evidence:**

Previous methods align video and language based on the cosine similarity between the average of frame-level [CLS] tokens in the video and the sentence-level [EOS] token. This limits the alignment to coarse semantic matching. In contrast, BDC-CLIP aligns the two modalities using Brownian Distance Covariance (BDC), which considers all visual and textual tokens. This approach captures fine-grained spatio-temporal cues crucial for action recognition.

**Essential References Not Discussed:**

None.

**Experimental Designs Or Analyses:**

The paper conducts experiments on five widely used action recognition datasets: Kinetics-400, Kinetics-600, HMDB-51, UCF-101, and SSv2. The model is evaluated across various downstream tasks, including zero-shot, few-shot, base-to-novel generalization, and fully-supervised settings. The proposed BDC-CLIP achieves state-of-the-art performance on these tasks.

**Methods And Evaluation Criteria:**

BDC-CLIP introduces two core components: (1) a video BDC adapter and (2) a text BDC adapter, which are aligned using Brownian Distance Correlation. (1) Video BDC Adapter: By leveraging all visual tokens (i.e., [CLS] and patch tokens), BDC-CLIP computes a BDC matrix as a frame-wise representation and designs a temporal attention mechanism to model frame-to-frame dynamics. (2) Text BDC Adapter: BDC-CLIP exploits all textual tokens (i.e., [EOS] and word tokens) to compute a BDC matrix as the text representation. Finally, BDC-CLIP aligns the video and text representations using Brownian Distance Correlation.

**Other Comments Or Suggestions:**

None.

**Other Strengths And Weaknesses:**

Strengths:
The paper is well-written and easy to understand.
The proposed method achieves state-of-the-art results on multiple datasets.

Weaknesses:
1, The main innovations of BDC-CLIP include:
(i) A video BDC adapter and a text BDC adapter, which serve as post-processing modules for the visual and text encoders to enhance alignment.
(ii) The use of Brownian Distance Covariance to strengthen alignment.
However, I have two concerns regarding the novelty of the paper:
(1) The idea of adding adapters after the CLIP encoder is not entirely new. Many existing works have explored similar techniques, including approaches related to relationships between sets of embeddings, global tokens, and local tokens. These are common strategies for processing CLIP tokens.
(2) The use of Brownian Distance Covariance primarily originates from image-based few-shot classification tasks and has been adapted for video action recognition in this work.
2, The motivation of this paper is not clearly articulated. The authors claim that previous methods rely on cosine similarity and only use global tokens. However, this is not entirely accurate, as many existing works already explore different similarity computations and incorporate local tokens. The paper does not sufficiently clarify this issue.
3, The paper states that BDC can capture both linear and nonlinear correlations, but it is unclear what exactly is meant by "linear" and "nonlinear" in this context. How do "linear" and "nonlinear" correlations correspond to specific methods in the proposed framework?
 There is a lack of experimental validation to demonstrate the advantage of capturing both linear and nonlinear correlations.

**Questions For Authors:**

The paper states that BDC can capture both linear and nonlinear correlations, but it is unclear what exactly is meant by "linear" and "nonlinear" in this context. How do "linear" and "nonlinear" correlations correspond to specific methods in the proposed framework? Which components in the method explicitly capture these correlations? There is a lack of experimental validation to demonstrate the advantage of capturing both linear and nonlinear correlations. It would be beneficial to include ablation studies or quantitative analyses to verify this claim.

**Relation To Broader Scientific Literature:**

None.

**Theoretical Claims:**

The idea of using Brownian Distance Covariance (BDC) for aligning video and language is theoretically reasonable.

---

> ### Author Rebuttal · Authors · 2025-04-01
>
> Dear Reviewer oB1H,
>
> We sincerely thank you for your constructive and insightful comments, especially your positive feedback that **"The paper is well-written and easy to understand"** and   **"The proposed method achieves state-of-the-art results on multiple datasets. "**
>
>
>
> > ### Q1: However, I have two concerns regarding the novelty of the paper: (1) The idea of adding adapters after the CLIP encoder is not entirely new. Many existing works have explored similar techniques, including approaches related to relationships between sets of embeddings, global tokens, and local tokens. These are common strategies for processing CLIP tokens. (2) The use of Brownian Distance Covariance primarily originates from image-based few-shot classification tasks and has been adapted for video action recognition in this work.
>
> Thank you for sharing these concerns.
>
> 1) **BDC Integration into CLIP.**
> As noted in Section 1 of our main paper, **our primary contribution is introducing BDC for vision-language alignment in foundation models like CLIP.** While past works such as DeepBDC [Xie et al., 2022] and BDC-Adapter [Zhang et al., 2023] apply BDC in single-modality, image-based few-shot classification, **none** address the cross-modal alignment challenge of video and text. Our approach goes beyond the predominantly used cosine similarity, demonstrating how BDC effectively captures statistical dependence between two distinct modalities. We also appreciate that **Reviewer GB1s recognized this novelty**, calling it the ***“novel integration of BDC into the CLIP framework.”***
>
> 2) **BDC-Based Temporal Adapter for Local and Global Tokens.**
> Our second contribution is **a temporal adapter tailored for BDC representation** that leverages both local (patch/word) tokens and global ([CLS]/[EOS]) tokens for adapting CLIP to video recognition. To the best of our knowledge, no prior work has applied BDC to modeling relationships among local tokens for CLIP-based video recognition. If such related work exists, we would welcome further references to ensure comprehensive coverage.
>
> Given these points, **we respectfully request a reconsideration of our paper’s novelty,** which combines BDC-based cross-modal alignment with a temporal adapter that captures fine-grained token interactions for video action recognition.
>
>
>
> > ### Q2: The motivation of this paper is not clearly articulated. The authors claim that previous methods rely on cosine similarity and only use global tokens. However, this is not entirely accurate, as many existing works already explore different similarity computations and incorporate local tokens. The paper does not sufficiently clarify this issue.
>
> Thank you for noting this issue.
>
> As we state in our Abstract (Lines 21-23) and Section 1 (Lines 57-71), most existing methods still rely on cosine similarity and focus on global tokens. To our best knowledge, **for CLIP-based video recognition,** *OST (Chen et al., 2024) is the only approach that departs from cosine similarity by using optimal transport*, but it aligns **frame-level [CLS] tokens with sentence-level [EOS] tokens rather than local (patch/word) tokens,** and its primary goal is to enhance textual descriptors. We have not encountered prior CLIP-based video recognition works that leverage local tokens for cross-modal alignment, but would appreciate any pointers if they exist. It is worth noting  **Reviewer LE28 affirmed that our *“motivation is clear and intuitive.”***
>
>
> > ### Q3: The paper states that BDC can capture both linear and nonlinear correlations, but it is unclear what exactly is meant by "linear" and "nonlinear" in this context. How do "linear" and "nonlinear" correlations correspond to specific methods in the proposed framework? There is a lack of experimental validation to demonstrate the advantage of capturing both linear and nonlinear correlations.
>
> Thank you for raising these points.
>
> **We address them in detail in (and kindly refer to) our response to Q1 from Reviewer GB1s,** where we explain *from the statistical perspective* the distinction between linear and nonlinear correlations (i.e., cosine similarity vs. BDC) and provide both theoretical rationale and qualitative evaluations (e.g., scatterplots and density contours and heatmaps) to illustrate BDC’s ability to capture more complex dependencies. Our extensive experiments in Section 4 further show that capturing complex dependencies--including linear and nonlinear correlations--has yielded superior performance, compared to strong baselines (i.e., VIFI-CLIP and TC-CLIP) based on cosine similarity that can only model linear correlations.

---

> > ### Comment · Reviewer_oB1H · 2025-04-07
> >
> > Thank the authors for the response. Some of my concerns have been addressed. However, there are still some issues remaining.
> >
> > For example, in the response to reviewer GB1s, the term "linear and nonlinear correlations" is not accurate enough. I cannot quite grasp what is meant by "nonlinear correlation"—my understanding is that it refers to the correlations between two set of tokens.
> >
> > Regarding the novelty of the paper, the integration of BDC into CLIP seems more like an engineering improvement rather than a theoretical innovation.
> >
> > I tend to give a borderline score (between 2 and 3), but since ICML only has a weak accept option, I have raised the score to 3.

---

> > > ### Author Response · Authors · 2025-04-07
> > >
> > > Dear Reviewer oB1H,
> > >
> > > Thank you for your thoughtful feedback and for raising your score. We’re pleased that some of your concerns have been addressed and appreciate the chance to clarify the remaining points.
> > >
> > > > ### Q1': For example, in the response to reviewer GB1s, the term "linear and nonlinear correlations" is not accurate enough. I cannot quite grasp what is meant by "nonlinear correlation"--my understanding is that it refers to the correlations between two set of tokens.
> > >
> > > We appreciate your perspective and would like to clarify that our intended meaning encompasses a broader concept.
> > >
> > > From a statistical perspective:
> > > * **Linear correlation** measures the degree to which two random variables (RVs) increase or decrease together in a linear manner—commonly quantified by the Pearson Correlation Coefficient (PCC). If the joint distribution is Gaussian, which features elliptical density contours, PCC fully characterizes their linear dependence.
> > > * **Nonlinear correlation** encompasses any statistical dependence beyond what PCC can capture, such as higher-order or non-monotonic relationships. In non-Gaussian distributions, these dependencies are often nonlinear.
> > >
> > > As formalized in (Zhelezniak et al., 2019), **token similarity can be measured through statistical correlations.** Specifically, each token embedding can be viewed as a sample of observations from a RV and cosine similarity (CS) is practically equivalent to PCC. Notably, CS/PCC performs optimally under linear or Gaussian assumptions. **However, our scatterplots clearly show that correlations between visual and textual tokens can be complex—nonlinear and non-Gaussian (refer to [Figure R1](https://anonymous.4open.science/r/rebuttal-D15F/Figure_R1.jpg)).** In such scenarios, CS/PCC, limited by its linear nature, inherently fails to capture these richer statistical dependencies effectively.
> > >
> > > Therefore, **existing CLIP adaptations** that rely on CS between global textual [EOS] token $\mathbf{w_0}$ and visual [CLS] token $\mathbf{p_0}$ across frames cannot effectively model nonlinear correlations. Our **BDC-CLIP** targets fine-grained vision-language alignment by measuring the similarity between all textual tokens $S_{\text{txt}} = \\{\mathbf{w_0}, \ldots, \mathbf{w_M}\\}$ and all visual tokens of one frame $S_{\text{img}} = \\{\mathbf{p_0}, \ldots, \mathbf{p_N}\\}$. Extending the framework of Zhelezniak et al. (2019), we view token embeddings $S$ as a collection of $d$-dimensional samples from a set of scalar RVs $R = \\{W_0, \ldots, W_M\\}$. Accordingly, **the similarity between $S_{\text{txt}}$ and $S_{\text{img}}$ can be naturally measured by the correlation between $R_{\text{txt}}$ and $R_{\text{img}}$.** BDC (Szekely & Rizzo, 2009)  provides a rigorous, general-purpose metric capable of capturing any form of statistical dependence—linear, nonlinear, Gaussian, or non-Gaussian—thus overcoming the inherent limitations of CS/PCC.
> > >
> > > > ### Q2': Regarding the novelty of the paper, the integration of BDC into CLIP seems more like an engineering improvement rather than a theoretical innovation.
> > >
> > > We recognize that integrating BDC into CLIP is not a theoretical innovation. However, our work ***introduces two significant contributions:***
> > >
> > > * **First Integration of BDC into Text & Visual Encoders for Vision–Language Alignment.** By replacing the ubiquitous cosine similarity with BDC in a foundation model like CLIP, we demonstrate the advantages of advanced statistical metrics for multimodal matching.
> > >
> > > * **A Temporal Attention Tailored for BDC Representations.**  We propose a custom temporal adapter that operates on BDC matrices using all tokens for video recognition, differing substantially from prior approaches that rely on temporal attention using only global tokens.
> > >
> > > Additionally, **our BDC-CLIP achieves state-of-the-art performance** across zero-shot, few-shot, base-to-novel, and fully supervised settings. We believe **these contributions are non-trivial to the community** and could spark broader  interest in exploring alternative metrics for multi-modality alignment in foundation models.
> > >
> > > Thank you once more for your time and valuable feedback. We trust these clarifications resolve your remaining concerns.
> > >
> > > Sincerely,
> > >
> > > The Authors

---

### Official Review · Reviewer_GB1s · 2025-03-12

**Overall Recommendation:** 3

**Summary:**

This paper introduces BDC-CLIP, a framework designed to adapt CLIP for video action recognition by using Brownian Distance Covariance (BDC). The authors claim that traditional methods, relying on cosine similarity on global tokens, lack the capacity to capture complex spatio-temporal relations in video data. Their proposed solution overcomes these limitations by leveraging BDC, which captures both linear and nonlinear relationships amongst all visual and textual tokens. Through extensive experimentation, the authors demonstrate state-of-the-art performance across various scenarios.

## update after rebuttal
I agree with the observations made by the other reviewers. The authors have clearly described their proposed method and provided detailed experimental results. However, the theoretical explanation and the visual evidence supporting the benefits of the proposed approach remain relatively weak.

That said, the rebuttal addressed many of my initial concerns, and I appreciate the authors’ efforts in clarifying their contributions and presenting additional analysis. If the authors can incorporate stronger theoretical insights and more comprehensive visualizations in the final version of paper, it would significantly strengthen the paper.

At this stage, I maintain my Weak Accept recommendation.

**Claims And Evidence:**

The primary claim of this paper revolves around the effectiveness of using Brownian Distance Covariance (BDC) for video-language alignment. This claim is supported by the ablation study in Table 6.

However, some important details lack sufficient evidence.
In the original statement:
“BDC can capture both linear and nonlinear correlations, enabling it to model the complex dependencies that exist between video and language embeddings.”

It is necessary to clarify: What exactly do linear and nonlinear correlations refer to in this context? How do they specifically manifest in video action recognition tasks? What qualitative or quantitative evaluation methods can be used to assess them?

**Essential References Not Discussed:**

Essential relates works are cited and discussed.

**Experimental Designs Or Analyses:**

The experimental designs make sense. They follow previous works.

**Methods And Evaluation Criteria:**

Yes, the method and evaluation criteria make sense.

**Other Comments Or Suggestions:**

Deeper analysis focused on interpretability, explaining why exactly BDC works better practically, is strongly recommended.

**Other Strengths And Weaknesses:**

Strengths:
1. Comprehensive evaluation compared with previous methods and ablation study
2. The novel integration of BDC into the CLIP framework

Weakness:
1. Marginal Empirical Improvements Relative to Computational Overhead:
The improvements shown over state-of-the-art baselines, while consistent, remain modest. Given the additional computational complexity and overhead introduced by integrating and computing BDC matrices, the incremental improvement may not be sufficient justification for adoption in real-world scenarios where computational efficiency is crucial.

2. Theoretical insights or deeper conceptual understanding are marginal or missing. First, a Preliminary Knowledge section is needed to introduce Brownian Distance Covariance (BDC). Following that, a thorough analysis of the theoretical motivation and insights should be provided, explaining the advantages of using BDC in video action recognition, how it captures non-linear relationships, and why non-linear relationship modeling is crucial for video action recognition tasks.

**Questions For Authors:**

None

**Relation To Broader Scientific Literature:**

The paper situates itself within recent trends of adapting image-language pretrained models for video understanding tasks. The use of BDC for video-language alignment is indeed novel in this context.

**Theoretical Claims:**

The authors leverage known theoretical foundations of Brownian distance covariance. However, their usage remains heuristic; no novel theoretical contributions or proofs are provided. Therefore, no theoretical analysis was performed or required for validation.

---

> ### Author Rebuttal · Authors · 2025-04-01
>
> Dear Reviewer GB1s,
>
> We sincerely thank you for providing constructive and insightful comments. Especially, we are grateful for your positive feedback including **"The *novel* integration of BDC into the CLIP framework "**, and **"Comprehensive evaluation compared with previous methods and ablation study."** alongside **"The authors demonstrate state-of-the-art performance across various scenarios."**
>
>
> > ### Q1: What exactly do linear and nonlinear correlations refer to in this context? How do they specifically manifest in video action recognition tasks? What qualitative or quantitative evaluation methods can be used to assess them?
>
> Thanks for raising these insightful concerns.
>
> 1) **Statistical perspective on linear & nonlinear correlations**
> * **Cosine Similarity (CS) $\approx $ Pearson Correlation Coefficient (PCC).** As formalized in (Zhelezniak et al., 2019), a token embedding can be viewed as a sample of observations from some scalar random variable (RV) and CS is practically equivalent to PCC that can capture only  linear correlations between RVs.
> * **Current CLIP-Based Methods.** Most existing adaptations apply CS on global tokens ([CLS]/[EOS]) across frames, only modeling linear dependencies between two scalar random variables (representing each token). This coarse alignment often overlooks more complex, fine-grained cues.
> * **BDC-CLIP.** We measure the similarity between the set of all textual tokens $S_{\text{txt}}=\\{\mathbf{w_{0}},\ldots, \mathbf{w_{M}} \\}$ and all visual tokens of one frame $S_{\text{img}}=\\{\mathbf{p_{0}},\ldots, \mathbf{p_{M}} \\}$. Extending from the statistical framework of Zhelezniak et al. (2019), we view textual embeddings $S_{\text{txt}}$ as a set of samples of $d$ observations from some theoretical set of scalar RVs $R=\\{W_0,\ldots, W_M\\}$. BDC can quantify all kinds of statistical dependency between two sets of RVs $R_{\text{txt}}$ and $R_{\text{img}}$ (Szekely & Rizzo,2009).
> 2) **Manifestation in Video Action Recognition**
> * Human actions exhibit rich contextual information across both spatial and temporal dimensions, involving dynamic interactions among people, objects, and the environment.
> * CS coarsely measures linear associations between the two modalities via global tokens, which  CS may miss crucial details unless they lie in a single linear dimension of the embedding space.
> * BDC, however, captures subtler interactions among fine-grained elements between language and vision, which can be highly nonlinear in the common embedding space.
> 3)	**Qualitative Evaluation**
> * We give **scatterplots and density contours** of some example visual vs. textual tokens; **kindly see [Figure R1](https://anonymous.4open.science/r/rebuttal-D15F/Figure_R1.jpg).** The plots show complex, nonlinear relations and clear non-Gaussian distributions, revealing nonlinear relation modeling is crucial and BDC can better capture these complex dependences.
> * We provide **heatmaps of example text-video frame pairs** for some action categories. These heatmaps highlight both key words (in shades of green) and salient spatial regions, evolving over time; **kindly see [Figure R2](https://anonymous.4open.science/r/rebuttal-D15F/Figure_R2.jpg).** The visualizations suggest BDC-CLIP can learn complex fine-grained multimodal context in space, time, and language.
>
> **Kindly refer to 2nd paragraph of Section 1 in our main paper for analysis on why nonlinear relation modeling is crucial for video action recognition.**
>
>
> > ### Q2: Marginal Empirical Improvements Relative to Computational Overhead.
>
> We appreciate the concern. We would like to emphasize that BDC-CLIP achieves substantial improvements over SOTA methods while maintaining a competitive computational profile. Specifically, BDC-CLIP significantly outperforms TC-CLIP: for zero-shot recognition, it achieves +4.7% on HMDB-51 and +1.5% on UCF-101; for base-to-novel task the gaps in light of HM are 2.6%, 2.0% and 2.1% on HMDB-51, UCF101 and SSv2, respectively. Meanwhile, BDC-CLIP uses fewer parameters (0.99x) and achieves higher throughput (1.12x), with only a slight increase in GFLOPs (1.04x). Overall, these gains justify BDC-CLIP’s use in real-world scenarios.
>
> > ### Q3: First, a Preliminary Knowledge section is needed to introduce Brownian Distance Covariance (BDC). Following that, a thorough analysis of the theoretical motivation and insights should be provided.
>
> Thanks for the thoughtful suggestions.
>
> In the revised paper, we will add a separate section, *in the Appendix (due to page limit in the main paper),* introducing background knowledge on BDC (Szekely & Rizzo, 2009). Then, we will highlight BDC’s theoretical advantages over PCC  that is practically equivalent to cosine similarity. Also, we will integrate the response to Q1, making clear what linear and nonlinear correlations mean in CLIP-based video recognition, meanwhile providing qualitative evaluation showcasing the importance of learning complex relations.

---

### Decision · Program_Chairs · 2025-05-01

**Decision:**

Accept (poster)

**Comment:**

This paper introduces a contrastive vision-language pre-training method for action recognition by using Brownian Distance Covariance for video-language alignment. Reviewers initially raised concerns regarding motivations and novelty, theoretical insights, and insufficient ablations of the proposed method.

After the rebuttal, the authors addressed most of the concerns, and all reviewers agreed to accept this paper, although some of the issues still remain (e.g., weak theoretical explanation and visual evidence). By taking the overall contributions and experimental results into consideration, AC believes most significant concerns have been alleviated and this paper is above the acceptance threshold.